# Uni-ControlNet: All-in-One Control to Text-to-Image Diffusion Models

**Shihao Zhao**[†]
The University of Hong Kong
shzhao@cs.hku.hk

**Dongdong Chen**[*]
Microsoft
cddlyf@gmail.com

**Yen-Chun Chen**
Microsoft
yen-chun.chen@microsoft.com

**Jianmin Bao**
Microsoft
jianmin.bao@microsoft.com

**Shaozhe Hao**
The University of Hong Kong
szhao@cs.hku.hk

**Lu Yuan**
Microsoft
luyuan@microsoft.com

**Kwan-Yee K. Wong**[*]
The University of Hong Kong
kykwong@cs.hku.hk

## Abstract

Text-to-Image diffusion models have made tremendous progress over the past two years, enabling the generation of highly realistic images based on open-domain text descriptions. However, despite their success, text descriptions often struggle to adequately convey detailed controls, even when composed of long and complex texts. Moreover, recent studies have also shown that these models face challenges in understanding such complex texts and generating the corresponding images. Therefore, there is a growing need to enable more control modes beyond text description. In this paper, we introduce Uni-ControlNet, a unified framework that allows for the simultaneous utilization of different local controls (e.g., edge maps, depth map, segmentation masks) and global controls (e.g., CLIP image embeddings) in a flexible and composable manner within one single model. Unlike existing methods, Uni-ControlNet only requires the fine-tuning of two additional adapters upon frozen pre-trained text-to-image diffusion models, eliminating the huge cost of training from scratch. Moreover, thanks to some dedicated adapter designs, Uni-ControlNet only necessitates a constant number (i.e., 2) of adapters, regardless of the number of local or global controls used. This not only reduces the fine-tuning costs and model size, making it more suitable for real-world deployment, but also facilitate composability of different conditions. Through both quantitative and qualitative comparisons, Uni-ControlNet demonstrates its superiority over existing methods in terms of controllability, generation quality and composability. Code is available at https://github.com/ShihaoZhaoZSH/Uni-ControlNet.

## 1 Introduction

In recent two years, diffusion models [1–10] have gained significant attention due to their remarkable performance in image synthesis tasks. Therefore, text-to-image (T2I) diffusion models [6, 7, 11–19] have emerged as a popular choice for synthesizing high-quality images based on textual inputs. By training on large-scale datasets with large models, these T2I diffusion models demonstrate exceptional ability in creating images that closely resemble the content described in text descriptions, and facilitate

---

[*]Corresponding Author, [†] Intern at Microsoft

37th Conference on Neural Information Processing Systems (NeurIPS 2023).

the connection between textual and visual domains. The substantially improved generation quality in capturing intricate texture details and complex relationships between objects, makes them highly suitable for various real-world applications, including but not limited to content creation, fashion design, and interior decoration.

However, text descriptions often prove to be either inefficient or insufficient to accurately convey detailed controls upon the final generation results, e.g., control the fine-grained semantic layout of multiple objects, not to mention the challenge in understanding complex text descriptions for such models. As a result, there is a growing need to incorporate more additional control modes (e.g., user-drawn sketch, semantic mask) alongside the text description into such T2I diffusion models. This necessity has sparked considerable interest from both academia and industry, as it broadens the scope of T2I generation from a singular function to a comprehensive system.

Very recently, there are some attempts [20–22] studying controllable T2I diffusion models. One representative work, Composer [20], explores the integration of multiple different control signals together with the text descriptions and train the model from scratch on billion-scale datasets. While the results are promising, it requires massive GPU resources and incurs huge training cost, making it unaffordable for many researchers in this field. Considering there are powerful pretrained T2I diffusion models (e.g., Stable Diffusion [6]) publicly available, ControlNet [21], GLIGEN [23] and T2I-Adapter [22] directly incorporate lightweight adapters (or extra modules) into frozen T2I diffusion models to enable additional condition signals. This makes fine-tuning more affordable. However, one drawback is that they need one independent adapter for each single condition, resulting in a linear increase in fine-tuning cost and model size along as the number of the control conditions grows, even though many conditions share similar characteristics. Additionally, this also makes composability among different conditions remains a formidable challenge.

In this paper, we propose Uni-ControlNet, a new framework that leverages lightweight adapters to enable precise controls over pre-trained T2I diffusion models. As shown in Table 1, Uni-ControlNet can not only handle different conditions within one single model but also supports composable control. By contrast, the existing methods fail to achieve this unified framework within one single model. Besides, even for those methods that support composite control, they perform poorly in terms of composability as illustrated in Section 4.

Unlike previous methods, Uni-ControlNet categorizes various conditions into two distinct groups: local conditions and global conditions. Accordingly, we only add two additional adapters, regardless of the number of local and global controls involved. This design choice not only significantly reduces both the whole fine-tuning cost and the model size, making it highly efficient for deployment, but also facilitates the composability of different conditions. To achieve this, we dedicatedly design the adapters for local and global controls. Specifically, for local controls, we introduce a multi-scale condition injection strategy that uses a shared local condition encoder adapter. This adapter first converts the local control signals into modulation signals, which are then used to modulate the incoming noise features. And for global controls, we employ another shared global condition encoder to convert them into conditional tokens, which are concatenated with text tokens to form the extended prompt and interacted with the incoming features via cross-attention mechanism. Interestingly, we find these two adapters can be separately trained without the need of additional joint training, while still supporting the composition of multiple control signals. This finding adds to the flexibility and ease of use provided by Uni-ControlNet.

By only training on 10 million text-image pairs with 1 epoch, our Uni-ControlNet demonstrates highly promising results in terms of fidelity and controllability. Figure 1 provides visual examples showcasing the effectiveness of Uni-ControlNet when using either one or multiple conditions. To gain further insights, we perform in-depth ablation analysis and compare our newly proposed adapter designs with those of ControlNet [21], GLIGEN [23] and T2I-Adapter [22]. The analysis results reveal the superiority of our adapter designs, emphasizing their enhanced performance over the counterparts offered by ControlNet, GLIGEN and T2I-Adapter.

## 2    Related Work

**Text-to-Image Generation**    is an emerging field that aims to generate realistic images from text descriptions. To address this challenging task, various approaches have been proposed in the past years. Early works [24–26] primarily adopted Generative Adversarial Networks (GANs) and were

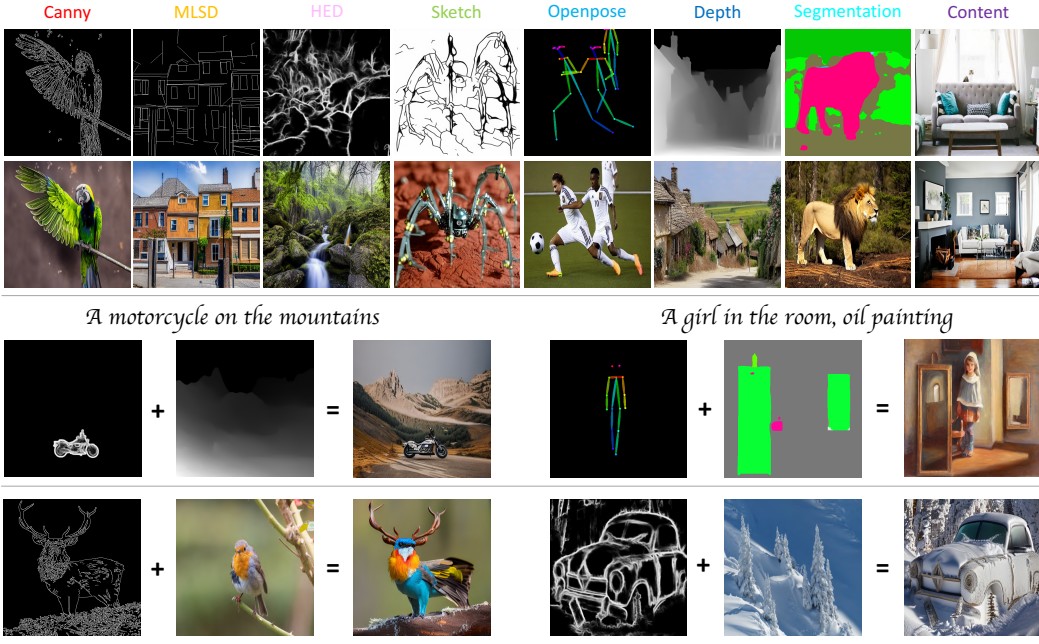

Figure 1: Visual results of our proposed Uni-ControlNet. The top and bottom two rows are results for single condition and multi-conditions respectively.

Table 1: Comparisons of different controllable diffusion models. $N$ is the number of conditions. We define the fine-tuning cost as the number of times the model needs to be fine-tuned on $N$ conditions. As Composer is trained from scratch, both fine-tuning cost and adapter number are not applicable. For T2I-Adapter, $(+1)$ indicates that further joint fine-tuning is required on the $N$-based adapters along with an additional fuser to achieve composable conditions.

|  | Fine-tuning | Composable Control | Fine-tuning Cost | Adapter Number |
|---|---|---|---|---|
| Composer | ✗ | ✔ | - | - |
| ControlNet | ✔ | ✔ | $N$ | $N$ |
| GLIGEN | ✔ | ✗ | $N$ | $N$ |
| T2I-Adapter | ✔ | ✔ | $N(+1)$ | $N(+1)$ |
| Uni-ControlNet (Ours) | ✔ | ✔ | 2 | 2 |

often trained on specific domains. However, they faced two key challenges, i.e., training instability and poor generalization ability to open-domain scenarios. Motivated by the success of GPT models [27–30], recent works [31–34] have explored the use of autoregressive models for text-to-image generation and train on web-scale image-text pairs, which start to show strong generation capability under the zero-shot setting for open-domain scenarios. Another approach is the diffusion models [6, 7, 11, 35–41], originally proposed by [1, 2]. Diffusion models comprise a forward process that gradually adds noise to natural images and a backward process that learns to denoise them back to generate clean output. They demonstrate stronger capability in modeling fine-grained structures and texture details compared to autoregressive models. Recently, vast variants of diffusion models have been developed, such as DALLE-2 [11], which uses one prior model and one decoder model to generate images from CLIP latent embeddings. Another phenomenal T2I diffusion model is Stable Diffusion (SD), which scaled up the latent diffusion model [6] with larger model and data scales, and made the pre-trained models publicly available. In this paper, we use SD as a base model and explore how to enable more control signals beyond the text description for pre-trained T2I diffusion models in an efficient and composable way.

**Controllable Diffusion Models** are designed to enable T2I diffusion models to accept more user controls for guiding the generation results. They have garnered increasing attention very recently.

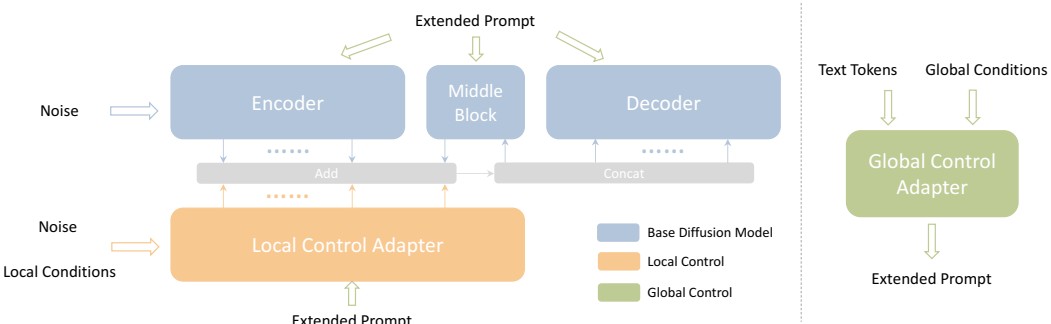

Figure 2: The overall framework of our proposed Uni-ControlNet.

Broadly speaking, there are two strategies for implementing controllable diffusion models: training from scratch [20] and fine-tuning lightweight adapters [21, 22] on frozen pretrained T2I diffusion models. In the case of training from scratch, Composer [20] trains one big diffusion model from scratch to achieve great controllability for both single and multi-conditions. It obtains remarkable generation quality but comes with huge training cost. In contrast, ControlNet [21], GLIGEN [23] and T2I-Adapter [22] propose to introduce lightweight adapters (or extra modules) into publicly available SD models. By only fine-tuning the adapters while keeping original SD models frozen, they significantly reduce the training cost and make it affordable for the research community. However, all the ControlNet, GLIGEN and T2I-Adapter utilize independent adapters for each condition, resulting in increased fine-tuning cost and model size when handling increased number of conditions. Moreover, GLIGEN does not support composite control over different conditions. And different adapters in Multi-ControlNet [21], a version of ControlNet that allow composite control, are isolated from one another, limiting their composability. By testing CoAdapter [22], which is jointly trained using different T2I-Adapters, we find that it also exhibits inadequate performance in generating composable conditions. Our proposed Uni-ControlNet follows the second line of fine-tuning adapters and is much less expensive than Composer, while addressing the above limitations of ControlNet, GLIGEN and T2I-Adapter. It groups conditions into two groups, i.e., local controls and global controls, and only requires two additional adapters accordingly. Thanks to our newly designed adapter structure, Uni-ControlNet is not only efficient in terms of training cost and model sizes, but also surpasses ControlNet, GLIGEN and T2I-Adapter in controllability and quality.

## 3 Method

### 3.1 Preliminary

A typical diffusion model involves two processes: a forward process which gradually adds small amounts of Gaussian noise onto the sample in $T$ steps, and a corresponding backward process containing learnable parameters to recover input images by estimating and eliminating the noise. In this paper, we use SD as our example base model to illustrate how to enable diverse controls with our Uni-ControlNet. SD incorporates the UNet-like structure [42] as its denoising model, which consists of an encoder, a middle block, and a decoder, with 12 corresponding blocks in each of the encoder and decoder modules. For brevity, we denote the encoder as $F$, the middle block as $M$, and the decoder as $G$, with $f_i$ and $g_i$ denoting the output of the $i$-th block in the encoder and decoder, and $m$ denoting the output of the middle block, respectively. It is important to note that, due to the adoption of skip connections in UNet, the input for the $i$-th block in the decoder is given by:

$$\begin{cases} concat(m, f_j) & where \quad i = 1, \quad i + j = 13. \\ concat(g_{i-1}, f_j) & where \quad 2 \leq i \leq 12, \quad i + j = 13. \end{cases} \tag{1}$$

Skip connections allow the decoder to directly utilize features from the encoder and thereby help minimize the information loss. In SD, cross-attention layers are employed to capture semantic information from the input text description. Here we use $Z$ to denote the incoming noise features and $y$ to denote text token embeddings encoded by the language encoder. The $Q, K, V$ in cross-attention can be expressed as:

$$Q = W_q(Z), K = W_k(y), V = W_v(y), \tag{2}$$

where $W_q, W_k$ and $W_v$ are projection matrices.

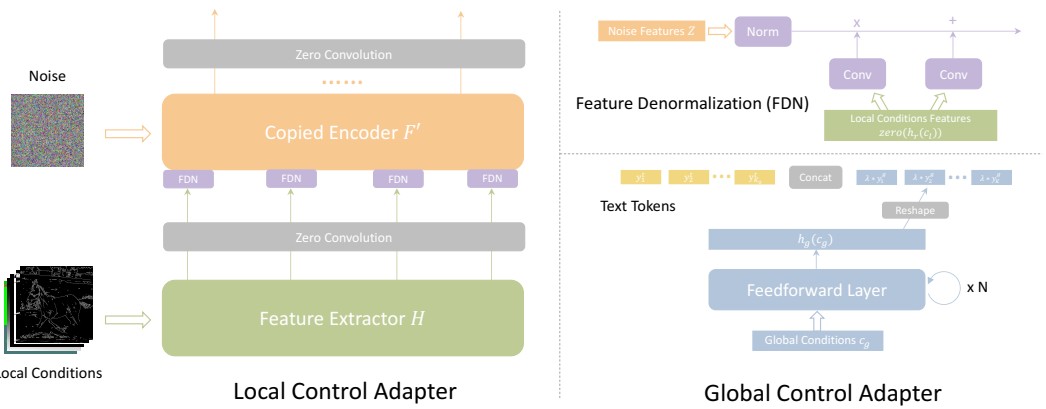

Figure 3: Details of the local and global control adapters.

## 3.2 Control Adapter

In this paper, we consider seven example local conditions, including Canny edge [43], MLSD edge [44], HED boundary [45], sketch [46, 47], Openpose [48], Midas depth [49], and segmentation mask [50]. We also consider one example global condition, i.e., global image embedding of one reference content image that is extracted from the CLIP image encoder [51]. This global condition goes beyond simple image features and provides a more nuanced understanding of the semantic content of the condition image. By employing both local and global conditions, we aim to provide a comprehensive control over the generation process. We show the overview of our pipeline in Figure 2, and the details of local control adapter and global control adapter are given in Figure 3.

**Local Control Adapter:** For our local control adapter, we have taken inspiration from ControlNet. Specifically, we fix the weights of SD and copy the structures and weights of the encoder and middle block, designated as $F'$ and $M'$ respectively. Thereafter, we incorporate the information from the local control adapter during the decoding process. To achieve it, we ensure that all other elements remain unchanged while modifying the input of the $i$-th block of the decoder as

$$\begin{cases} concat(m + m', f_j + zero(f_j')) & where \quad i = 1, \quad i + j = 13. \\ concat(g_{i-1}, f_j + zero(f_j')) & where \quad 2 \le i \le 12, \quad i + j = 13. \end{cases} \quad (3)$$

where $zero$ represents one zero convolutional layer whose weights increase from zero to gradually integrate control information into the main SD model. In contrast to ControlNet that adds the conditions directly to the input noise and sends them to the copied encoder, we opt for a multi-scale condition injection strategy. Our approach involves injecting the condition information at all resolutions. In detail, we first concatenate different local conditions along the channel dimension and then use a feature extractor $H$ (stacked convolutional layers) to extract condition features at different resolutions. Subsequently, we select the first block of each resolution (i.e., $64 \times 64, 32 \times 32, 16 \times 16, 8 \times 8$) in the copied encoder (i.e., the Copied Encoder in Figure 3) for condition injection. For the injection module, we take the inspiration from SPADE [52] and implement Feature Denormalization (FDN) that uses the condition features to modulate the normalized (i.e.,$norm(\cdot)$) input noise features:

$$FDN_r(Z_r, c_l) = norm(Z_r) \cdot (1 + conv_\gamma(zero(h_r(c_l)))) + conv_\beta(zero(h_r(c_l))), \quad (4)$$

where $Z_r$ denotes noise features at resolution $r$, $c_l$ is the concatenated local conditions, $h_r$ represents the output of the feature extractor $H$ at resolution $r$, and $conv_\gamma$ and $conv_\beta$ refer to learnable convolutional layers that convert condition features into spatial-sensitive scale and shift modulation coefficients. We will ablate different local feature injection strategies in following sections.

**Global Control Adapter:** For global controls, we use the image embedding of one condition image extracted from CLIP image encoder as the example. Inspired by the fact that the text description in T2I diffusion models can be also viewed as one kind of global control without explicit spatial guidance, we project the global control signals into condition embeddings by using a condition encoder $h_g$. The condition encoder consists of stacked feedforward layers, which aligns the global control signals with the text embeddings in SD. Next, we reshape the projected condition embeddings into $K$ global tokens ($K = 4$ by default) and concatenate them with the original $K_0$ text tokens

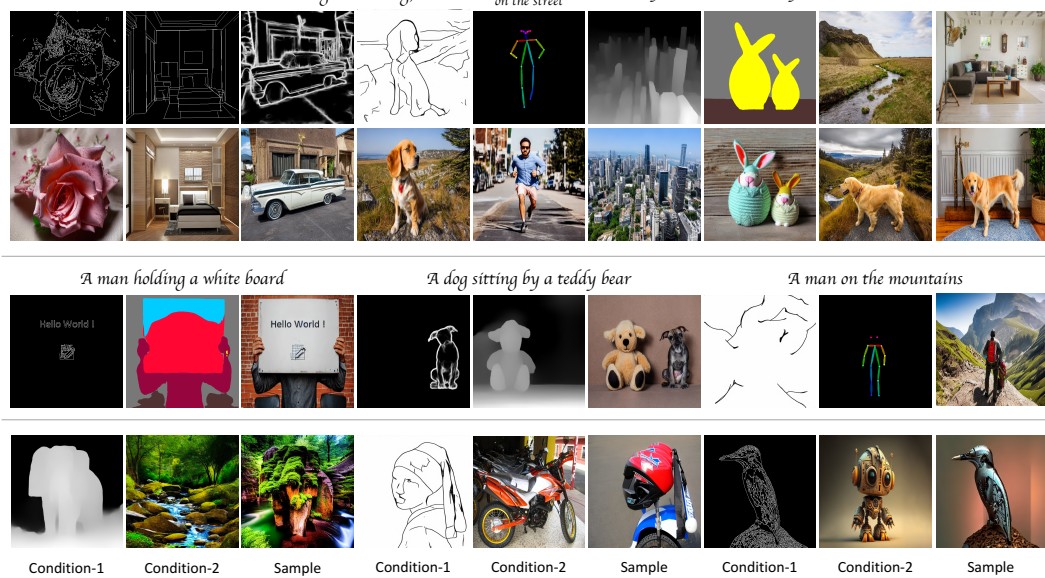

Figure 4: More visual results of Uni-ControlNet. The top two rows show results of a single condition, with columns 1-7 for local conditions and columns 8-9 for global condition. 3rd row shows the results of combining two local conditions, while row 4-th shows the results of integrating a local condition with a global condition. There is no text prompt for the examples in 4-th row.

to create an extended prompt $y_{ext}$ (total token number is $K + K_0$) which serves as the input to all cross-attention layers in both main SD model and control adapters:

$$y_{ext} = [y_1^t, y_2^t, ..., y_{K_0}^t, \lambda * y_1^g, \lambda * y_2^g, ..., \lambda * y_K^g], \;\; where \;\; y_i^g = h_g(c_g)[(i-1) \cdot d \sim i \cdot d], i \in [1, K] \tag{5}$$

where $y^t$ and $y^g$ represent the original text tokens and global condition tokens respectively, and $\lambda$ is a hyper-parameter that controls the weight of the global condition. $c_g$ denotes the global condition and $d$ is the dimension of text token embedding. $h_g(\cdot)[i_s \sim i_e]$ represents the sub-tensor of $h_g(\cdot)$ that contains elements from the $i_s$-th to the $i_e$-th positions. Finally, the $Q, K, V$ cross-attention operation in all cross-attention layers is changed to:

$$Q = W_q(Z), K = W_k(y_{ext}), V = W_v(y_{ext}), \tag{6}$$

### 3.3 Training Strategy

As the local control signals and global control signals often contain different amounts of condition information, we empirically find that directly joint fine-tuning these two types of adapters will produce poor controllable generation performance. Therefore, we opt to fine-tune these two types of adapters separately so that both of them can be sufficiently trained and contribute effectively to the final generation results. When fine-tuning each adapter, we employ a predefined probability to randomly dropout each condition, along with an additional probability to deliberately keep or drop all conditions. For the dropped conditions, we set the value of the corresponding input channels to 0. This can facilitate the model to learn generating the results based on one or multiple conditions simultaneously. Interestingly, by directly integrating these two separately trained adapters during inference, our Uni-ControlNet can already well combine global and local conditions together in a composable way, without the need of further joint fine-tuning. In Section 4.3, we will provide more detailed analysis about different fine-tuning strategies.

## 4 Experiments

**Implementation Details.** To fine-tune our model, we randomly sample 10 million text-image pairs from the LAION dataset [53] and fine-tune Uni-ControlNet for 1 epoch. We use the AdamW optimizer [54] with a learning rate of $1 \times 10^{-5}$ and resize the input images and local condition maps to $512 \times 512$. As described, the local and global control adapters are fine-tuned separately by

default. During inference, we merge the two adapters and adopt DDIM [55] for sampling, with the number of time steps set to 50 and the classifier free guidance scale [56] set to 7.5. During training, the hyper-parameter $\lambda$ in Equation 6 is with a fixed value 1. At inference time, when there is no text prompt, $\lambda$ remains at 1, while when there is a text prompt, the value is adjusted to around 0.75, depending on the intented weight between the text and global condition. As explained in Section 3.2, we employ 7 local conditions (Canny edge, MLSD edge, HED boundary, sketch, Openpose, Midas depth, and segmentation mask) and 1 global control condition (CLIP image embeddings) for control. As annotating a sketch dataset can be challenging, in our experiment, we initially obtain the HED boundary [45] of an image and subsequently utilize a sketch simplification method [46, 47] to generate the sketch for the training sample. Regarding the pose condition, as not all images in the dataset include humans, we opt to not drop the pose condition during training to ensure that the pose condition is fully trained. Detailed structures of global and local condition adapters can be found in the appendix.

## 4.1   Controllable Generation Results

In Figure 4, we provide more controllable generation results of Uni-ControlNet in both single and multi-condition setups. Notably, for visualization purposes, we use the original condition images to denote their CLIP image embeddings. It can be seen that our Uni-ControlNet can produce very promising results in terms of both controllability and generation fidelity. For example, in the case of a single sketch condition with the text prompt "Dog, wild" (rows 1-2, column 4), the resulting image accurately depicts a vivid dog and a background of grass and trees that align well with the given sketch condition. Similarly, when presented with the global CLIP image embedding conditions with the prompt "Golden retriever" (rows 1-2, columns 8-9), our model can seamlessly change the background of the dog from the wild to a room. Moreover, our model also handles multi-condition settings well, as demonstrated in the example of "A man on the mountains" (row 3, columns 7-9), where the combination of a sketch and a pose produces a cohesive and detailed image of a man on a mountainside. When presented with a local depth map and global CLIP image embeddings without any prompt (row 4, columns 1-3), our model produces an image of a forest, taking the contour of an elephant, which harmonizes with both the depth map and the content of the source global image.

## 4.2   Comparison with Existing Methods

Here we compare our Uni-ControlNet with ControlNet (Multi-ControlNet) [21], GLIGEN [23] and T2I-Adapter (CoAdapter) [22]. Since Composer [20] is not open-sourced and trained from scratch, we do not include it in comparisons.

**Quantitative Comparison:** For quantitative evaluation, we use the validation set of COCO2017 [57] at a resolution of $512 \times 512$. Since this set contains 5k images, and each image has multiple captions, we randomly select one caption per image resulting in 5k generated images for our evaluation. It is important to note that for quantitative comparison, we limit our testing to different single conditions only. Additionally, we use Style\Content to represent the global condition as there are different settings in the ControlNet, GLIGEN and T2I-Adapter. For the ControlNet, the content condition refers to the content shuffle in ControlNet-V1.1. As T2I-Adapter does not take the MLSD and HED conditions into account, it has no results for MLSD and HED. Similarly, GLIGEN does not consider the MLSD and sketch conditions, resulting in the absence of results for MLSD and sketch.

To evaluate the generation quality, We report the FID [58] in Table 2. We can find that our model reveals superior performance across most conditions quantitatively compared to existing approaches. We also use quantitative metrics to assess the controllability. We employed the following metrics for single-condition generation:

- SSIM (Structural Similarity) for Canny, HED, MLSD, and sketch conditions,
- mAP (mean Average Precision) based on OKS (Object Keypoint Similarity) for pose condition,
- MSE (Mean Squared Error) for depth map,
- mIoU (Mean Intersection over Union) for segmentation map,
- CLIP score for content condition.

Table 2: FID on different controllable diffusion models. The best results are in **bold**.

|  | Canny | MLSD | HED | Sketch | Pose | Depth | Segmentation | Style\Content |
|---|---|---|---|---|---|---|---|---|
| ControlNet | 18.90 | 31.36 | 26.59 | 22.19 | 27.84 | 21.25 | **23.08** | 31.17 |
| GLIGEN | 24.74 | - | 28.57 | - | **24.57** | 21.46 | 27.39 | 25.12 |
| T2I-Adapter | 18.98 | - | - | **18.83** | 29.57 | 21.35 | 23.84 | 28.86 |
| Ours | **17.79** | **26.18** | **17.86** | 20.11 | 26.61 | **21.20** | 23.40 | **23.98** |

Table 3: Quantitative evaluation of the controllability. The best results are in **bold**.

|  | Canny (SSIM) | MLSD (SSIM) | HED (SSIM) | Sketch (SSIM) | Pose (mAP) | Depth (MSE) | Segmentation (mIoU) | Style\Content (CLIP Score) |
|---|---|---|---|---|---|---|---|---|
| ControlNet | 0.4828 | **0.7455** | 0.4719 | 0.3657 | 0.4359 | **87.57** | **0.4431** | 0.6765 |
| GLIGEN | 0.4226 | - | 0.4015 | - | 0.1677 | 88.22 | 0.2557 | 0.7458 |
| T2I-Adapter | 0.4422 | - | - | 0.5148 | **0.5283** | 89.82 | 0.2406 | 0.7078 |
| Ours | **0.4911** | 0.6773 | **0.5197** | **0.5923** | 0.2164 | 91.05 | 0.3160 | **0.7753** |

To calculate these metrics, we compare the extracted conditions from the natural image (the ground truth) and the corresponding generated image. And we report the results in Table 3. Our method outperforms other baseline methods in 4 out of 8 evaluation metrics. Notably, ControlNet achieves the best performance in 3 out of 8 metrics, while T2I-Adapter only excels in 1 out of 8 metrics. However, it should be noted that all of ControlNet, GLIGEN and T2I-Adapter employ different models for different conditions, allowing each model to be well-trained for its corresponding condition. In contrast, we only use a single model and achieved even overall superior results.

To provide a more comprehensive comparison of various controllable models, we also include a comparison on CLIP score in Table 7 and present the results of user studies in Section G in the appendix.

**Qualitative Comparison:** We further provide qualitative comparison of single and composed multi-conditions in Figure 5 and Figure 6 respectively. For single conditions, as GLIGEN not considering the sketch condition, we use GLIGEN's results on the HED boundary as the second case in the first row for showcase. We find that our Uni-ControlNet, ControlNet, GLIGEN and T2I-Adapter can all perform overall well in single condition setting, and our results show slightly better alignments with input conditions. Notably, we only fine-tune 2 adapters for all conditions, whereas ControlNet, GLIGEN and T2I-Adapter fine-tune eight adapters for eight different single conditions.

Since GLIGEN does not support composed multi-conditions, we only compare Uni-ControlNet with Multi-ControlNet and CoAdapter under the multi-condition setting. As shown in Figure 6, Multi-ControlNet and CoAdapter show poorer composability when dealing with two local conditions, e.g., missing the podium in the first example and no car in the second example. In contrast, our model can fuse the two conditions much better. As for composing a local condition with a global condition, Multi-ControlNet is also not that good as shown in the second row in Figure 6. And CoAdapter performs okay in the case of combing a sketch of a cup and a global condition of a cat. However, when the two conditions are not that related, e.g., the example where there is a Canny edge of a Minion and a global condition of a bus in London, the image generated by CoAdapter appears to be unrealistic and the two elements are not well integrated. And our model effectively creates a Minion-shaped bus with car windows and vivid background.

## 4.3 Ablation Analysis

For ablation study, we fine-tune our model using a smaller dataset for resource consideration. In detail, we utilize the 1 million subset of the 10 million dataset and fine-tune a single epoch, while keeping all other settings unchanged.

**Condition Injection Strategy:** For local conditions, we compare our proposed injection method with two other strategies. The first strategy is to directly use SPADE to inject the conditions, which involves resizing the conditions to the corresponding resolutions using interpolation. We call this Injection-S1. The second strategy is similar to Composer, ControlNet and T2I-Adapter, where the conditions are only sent to the adapter or the main model at the input layer, which we refer to as Injection-S2. When using these two strategies, all other parts of our method will remain unchanged. We follow the setting in Section 4.2 and evaluate the FID on different local condition injection strategies. The quantitative

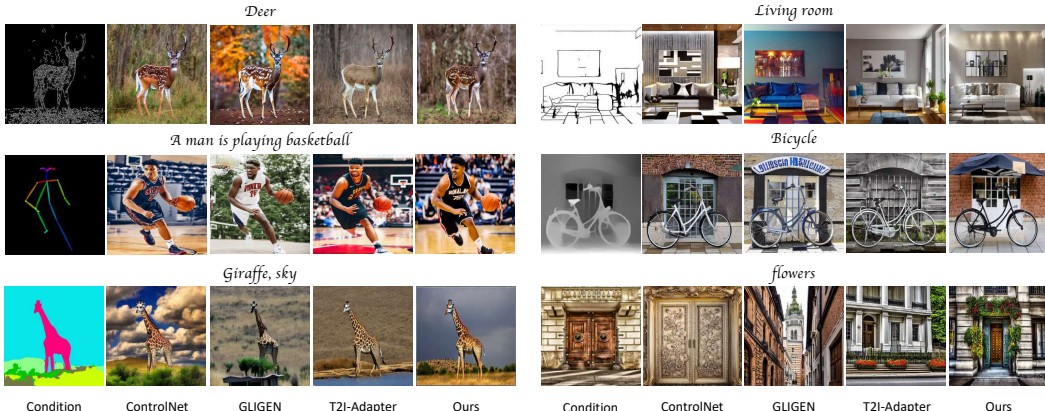

Figure 5: Comparison of existing controllable diffusion models on different single conditions.

Table 4: FID on condition injection methods and training strategies. The best results are in **bold**.

|  | Canny | MLSD | HED | Sketch | Openpose | Depth | Segmentation | Content |
|---|---|---|---|---|---|---|---|---|
| Injection-S1 | 25.89 | 27.22 | 22.48 | 23.51 | 27.89 | 24.71 | 26.25 | - |
| Injection-S2 | 22.22 | 27.08 | 21.94 | 22.74 | **26.56** | 24.21 | 24.42 | - |
| Injection-S3 | - | - | - | - | - | - | - | 27.06 |
| Training-S1 | 21.21 | 27.20 | 20.78 | 23.22 | 27.83 | 25.01 | 24.99 | 28.51 |
| Training-S2 | 18.80 | **26.40** | 19.12 | 20.91 | 27.17 | **21.59** | 23.93 | **24.84** |
| Ours | **18.24** | 26.91 | **18.61** | **20.32** | 27.76 | 21.97 | **23.51** | 24.86 |

and qualitative results are presented in Table 4 and the upper part of Figure 7. The quantitative results show our proposed condition injection strategy performs better under most settings. For the visual results, we observe that for Injection-S1, the alignment with the conditions is poor. This may be because direct interpolation significantly destroys condition information. As for Injection-S2, it renders unsatisfactory results for composite control. For instance, in the "An elephant in the temple" case, the lanterns on the top of the image are not accurately aligned with depth condition. Moreover, the composite results are not as harmonious as ours. For example, in the "Gorilla wearing glasses" case, the gorilla's eyes and glasses are not well-merged. This may be because that if the condition information is only provided in the input layer of the adapter, the model may lose some information of the conditions in deeper layers, leading to poor alignment among the combined controls. In contrast, our proposed FDN employs a multi-scale injection strategy that provides condition information at different levels, resulting in richer condition information. Furthermore, our feature extractor projects the conditions to the corresponding latent spaces of different layers, which allows for better alignment between the conditions and noise features.

For the global condition, we compare our method to one way in which we only add the global condition into control adapter but not the main SD model, and we denote this injection strategy as Injection-S3. As shown in the lower part of Figure 7, without using the extended prompt in the main SD model, this method cannot inject the global condition into the final generated results.

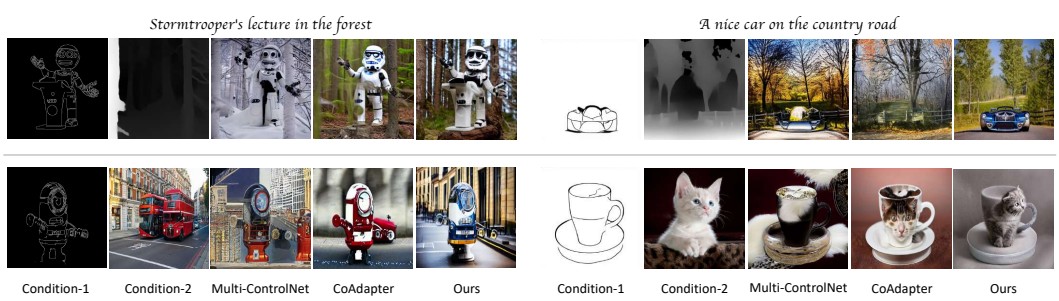

Figure 6: Comparison of different controllable diffusion models on composable multi-conditions.

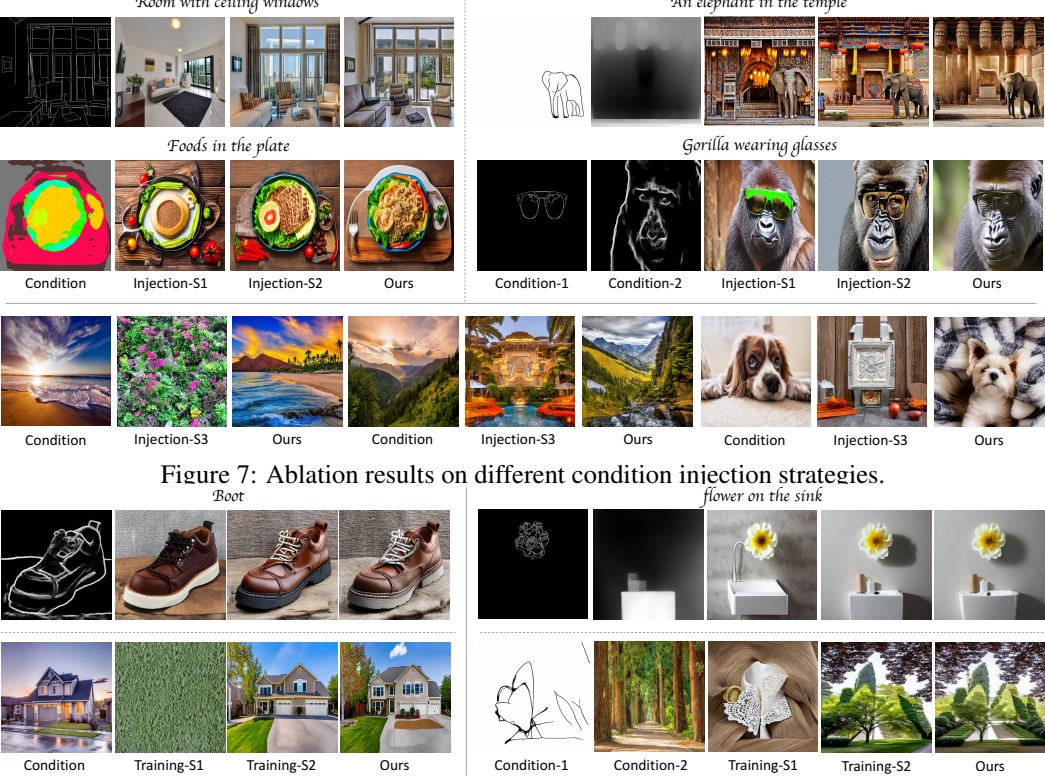

Figure 7: Ablation results on different condition injection strategies.

Figure 8: Ablation results on different training strategies.

**Training Strategy:** As above mentioned, we fine-tune the local and global control adapters separately and merge them at inference without any further joint fine-tuning by default. Here, we also investigate two alternative training strategies: 1) joint fine-tuning together ("Train-S1"), where we fine-tune both adapters together from scratch; 2) further joint fine-tuning after separate fine-tuning ("Train-S2"), where we further fine-tune the adapters together after separate fine-tuning.

The quantitative FID results are shown in Table 4. We find that our default strategy and Training-S2 perform much better consistently than Training-S1, but further joint fine-tuning in Train-S2 does not bring obvious performance gain in most cases. Some visual results are given in Figure 8. Note that, in order to better assess the controllability of the global condition, we do not provide text prompts for the cases with global condition. As we described before, the reason why Training-S1 gets poor controllability on the global condition is that the global control adapter does not learn as much as local adapter even equally treated during joint fine-tuning. One possible explanation is that the local conditions often contain more rich guidance information than global conditions, leading the model to pay less attention to the global condition.

## 5   Conclusion and Social Impact

In this paper, we propose Uni-ControlNet, a new solution that enhances the capabilities of text-to-image diffusion models by enabling efficient integration of diverse local and global controls. With better adapter designs, our Uni-ControlNet only requires two adapters for different conditions while existing methods often require independent adapters for each condition. The new design of Uni-ControlNet not only saves both fine-tuning cost and model size, but also facilitates composability, allowing for the simultaneous utilization of multiple conditions. Extensive experiments validate the effectiveness of Uni-ControlNet, showcasing its improved controllability, generation fidelity, and composability. While our system empowers artists, designers, and content creators to realize their creative visions with precise control, it is crucial to acknowledge the potential negative social impact that can arise from misuse or abuse, similar to other image generation and editing AI models. To address these concerns, responsible deployment practices, ethical regulations, and the inclusion of special flags in generated images to enhance transparency are vital steps towards responsible usage.

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

# A    The Weight of the Global Condition

In the global control module, we have implemented a hyper-parameter $\lambda$. This hyper-parameter plays a role in determining the influence of the global condition while concatenating the projected global condition to the text. Illustrating the effect of varying $\lambda$, we present two representative visualization results in Figure 9. It can be seen that the value of $\lambda$ plays a significant role in displaying the elements of the global conditions in the generated images. As the value of $\lambda$ increases, the global condition takes precedence over the original text content, leading to a decrease in the influence of the text prompt on the results. For instance, in the first case, it appears that the forest has experienced a reduction in coverage area as compared to an increase in the number of houses with an increase in the value of $\lambda$. Similarly, in the second case, it is evident that the city is shrinking while the desert's coverage is expanding with the rise of $\lambda$ value. In the real-world applications, we can adjust the hyper-parameter $\lambda$ to generate our desired results with flexibility.

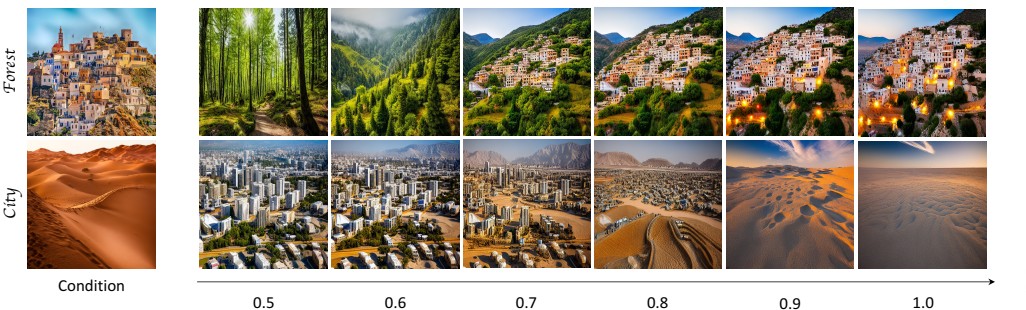

Figure 9: The effect of the hyper-parameter $\lambda$. On the left side are the textual prompts and global conditions provided. On the right side are the images generated under increased $\lambda$ values.

# B    Condition Conflicts

Since our Uni-ControlNet can support multiple conditions simultaneously, we are curious about its behavior when providing multiple conflicting conditions. Need to that, this is very rare in the real-world applications, and this experiment is just for the analysis purpose. For example, we consider the case of providing the model with two local conditions that are fundamentally incompatible, such as the conditions of two dogs shown in Figure 10. Through this experiment, we can possibly evaluate the relative importance of each condition and learn how Uni-ControlNet resolves conflicts, which may help us design more robust integration of conditions that can adequately handle diverse and ambiguous situations.

To provide a comprehensive analysis of different condition compositions, we have assigned each condition in the first column a number 1 and each condition in the first row a number 2. This allows us to refer to the dog in the first row as dog-1 and the dog in the first line as dog-2. The results depicted in Figure 10 demonstrate that HED is the most powerful condition, with generated images closely following the HED boundary when depicting text. Other conditions can only influence areas that do not overlap. For instance, when we combine the HED boundary of dog-2 with the Canny edge map of dog-1, the resulting image adopts the HED boundary of dog-2 but recognizes the head of dog-1 as a small element positioned near the head of dog-2. Similarly, when dog-2's HED boundary is combined with the sketch of dog-1, the model fails to identify the head of dog-1 even though it does not conflict with the HED boundary. Among the other conditions, the Canny edge map is the next most powerful, followed by the sketch, depth, MLSD, and segmentation map. The Openpose condition is the weakest, whereby the model generally disregards it in the event of a conflict. Only when combined with the segmentation map, the Openpose condition produces recognizable human elements. For better visualization, we have reordered the conditions based on their strength, which implies that the upper and left conditions have greater influence.

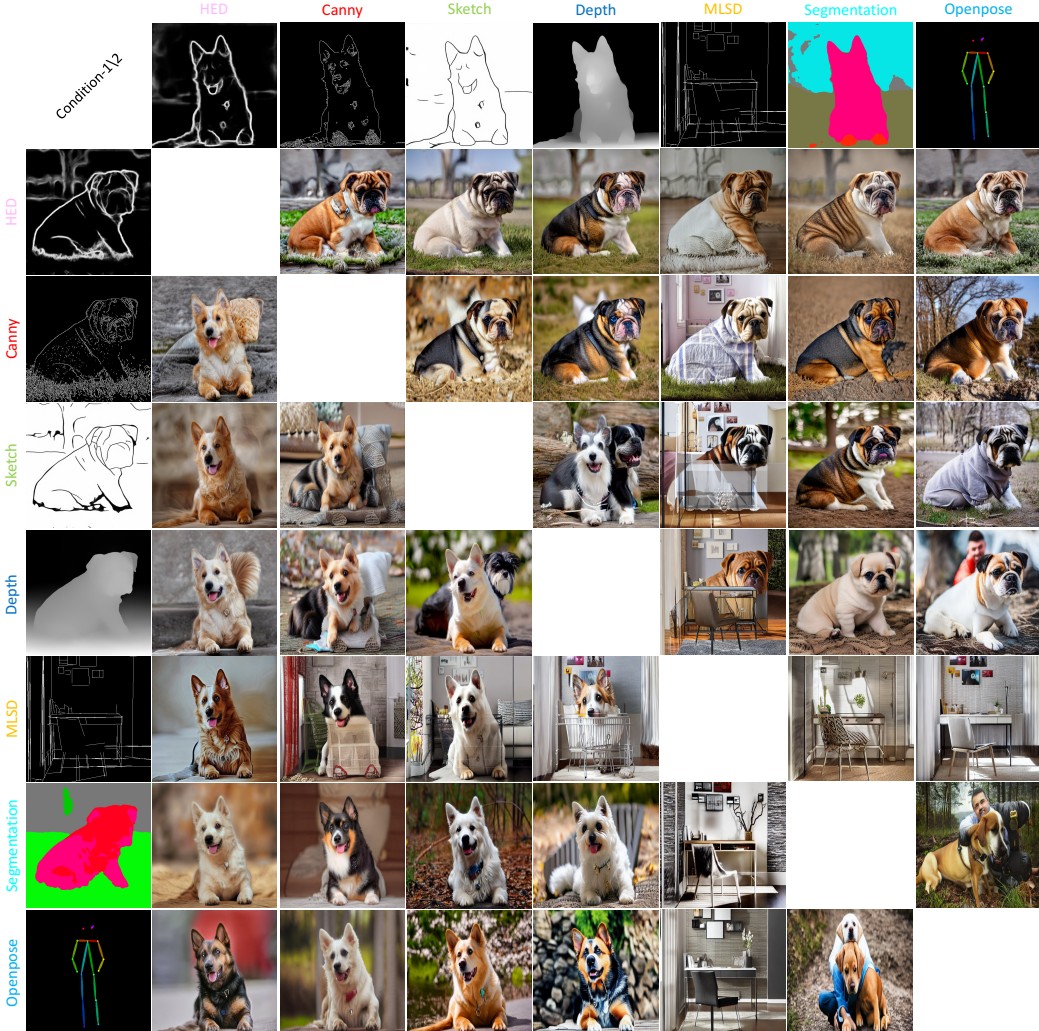

Figure 10: Study for the cases where composed conditions contradict each other. We choose the detection results of a room as the condition for MLSD and a man as the condition for Openpose. To test other conditions, we select two different dogs, which allows us to observe the model's output when given different dog-shaped conditions. We use "room" as the prompt for MLSD, "man" for Openpose, and "dog" for other conditions. When combining two types of conditions, we integrate their prompts, such as "dog", "dog and room", and "dog and man".

## C Hand-drawn Sketches

For the sketch condition, as mentioned in the main paper, we first get the HED boundary [45] of the training images. Then, we employ a sketch simplification method [46, 47] to generate the sketches for the model training. One question is, how does our model perform on hand-drawn sketches? We show the results of our model on hand-drawn sketches in Figure 11. We can find that although there are distribution gaps between the hand-drawn sketches and the model-generated sketches, our model can handle hand-drawn sketches pretty well.

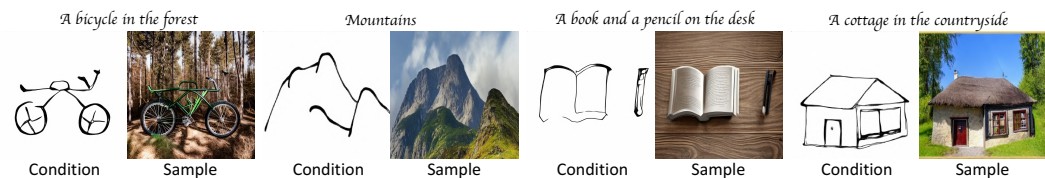

Figure 11: Visualization results on hand-drawn sketches.

## D Extension to New Conditions

To extend a trained Uni-ControlNet to support new conditions, we conducted an experiment in two steps for comparison and analysis purpose. Firstly, we train a local adapter specific to N conditions. Next, we introduce a new type of condition and extend the trained adapter to (N+1) conditions. The adaptation process involved modifying the input channel of the Uni-ControlNet's first convolutional layer within the feature extractor. Then, we try to retrain the local adapter with 4 different retraining strategies (R1-4) to accommodate the new conditions:

- Retraining the entire feature extractor (R1),
- Only retraining the pre-feature extractor, which is the part that projects the condition from resolution 512 to 64 (R2),
- Only retraining the first convolutional layer in the feature extractor (R3),
- Without retraining, i.e., random initialization of the first convolutional layer in the feature extractor (R4).

During the retraining process, we ensure that the weights of the copied encoder in the local adapter remain fixed. We utilize a training dataset of 300k samples for the retraining. We show the extension from [MLSD + HED + Sketch + OpenPose + Depth + Seg] to [MLSD + HED + Sketch + OpenPose + Depth + Seg + Canny]. The results of this extension process are presented in Figure 12. We surprisingly observe that retraining solely the first convolutional layer in the feature extractor can already adequately enable the Uni-ControlNet to handle the newly added conditions.

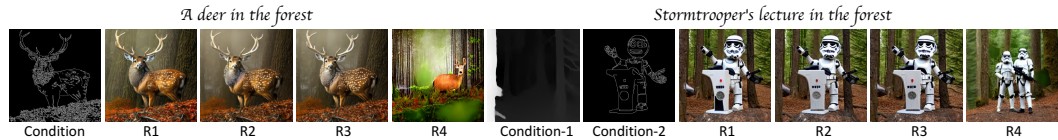

Figure 12: Study for extending a trained Uni-ControlNet to newly added conditions.

## E Composite Control of Conditions with the Same Type

In real-world applications, users can actually composite two/multiple conditions of the same type easily before feeding them to the model, e.g., draw the sketch of multiple objects in one canvas. However, how to achieve composite control of two conditions with the same type is an interesting research point. We try one simple strategy called "Uni-Channels". Specifically, we augment the input

by adding three extra condition channels. For instance, if the original inputs had 21 channels (3 for each condition, totaling 7 local conditions), with Uni-Channels, we now have 21 + 3 channels for the inputs.

During training, we feed the Uni-Channels with randomly selected types of conditions of the input natural images. We observe that, as the shared extra condition channels for different condition types, Uni-Channels can perform well for two-condition composition of the same condition type. The visualization results are depicted in Figure 13.

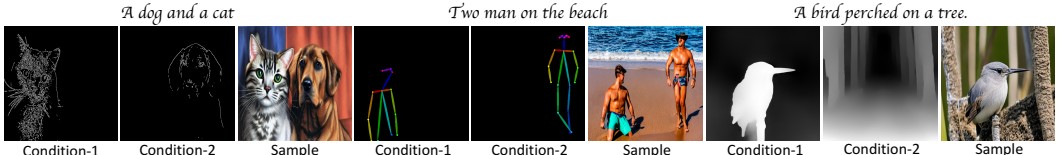

Figure 13: Visualization results of composite control of two conditions with the same type.

# F  Comparison with Stable Diffusion 2.1

We compare our method with two Stable Diffusion models, Stable Diffusion 2 - depth ("SD2-depth") and Stable Diffusion 2 - unclip ("SD2-unclip") which support the inputs of depth map and reference image respectively. The visualization results are shown in Figure 14. Additionally, we provide the quantitative results in Table 5 and Table 6.

Table 5: FID on Uni-ControlNet and Stable Diffusion 2.1. The best results are in bold.

| FID | Depth | Content |
|---|---|---|
| SD2-depth | **17.76** | - |
| SD2-unclip | - | 24.12 |
| Ours | 21.20 | **23.98** |

Table 6: CLIP score on Uni-ControlNet and Stable Diffusion 2.1. The best results are in bold.

| CLIP Score | Depth | Content |
|---|---|---|
| SD2-depth | 0.2516 | - |
| SD2-unclip | - | **0.2497** |
| Ours | **0.2561** | 0.2402 |

It is important to note that for SD2-depth and SD2-unclip, the whole model is fine-tuned to learn the depth map or the reference images instead of only fine-tuning adapters, which is the key factor contributing to their great performance. Additionally, when compared to other controllable diffusion models like ControlNet, GLIGEN and T2I-Adapter, SD2-depth and SD2-unclip outperform them, as demonstrated in Table 2 and Table 7.

# G  More Quantitative Results

**CLIP Score:**   Besides FID, we also test CLIP score for comparing different controllable diffusion models, and ablating condition injections methods and training strategies. We follow the settings in the Section 4.2 and Section 4.3 in the main paper. The results are shown in the Table 7 and Table 8 respectively. Our model demonstrates superior performance quantitatively across most conditions when compared to existing controllable diffusion models. Moreover, for different condition injection methods and training strategies, our method, along with Training-S2, consistently outperforms other strategies. However, joint fine-tuning in Training-S2 does not yield obvious performance gains in most cases. These finds are consistent with those presented in the Section 4.2 and Section 4.3 of the main paper.

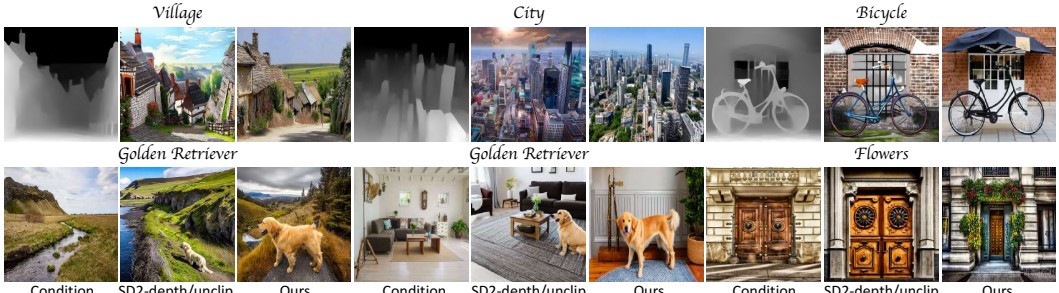

*Village*      *City*      *Bicycle*

*Golden Retriever*      *Golden Retriever*      *Flowers*

Condition    SD2-depth/unclip    Ours     Condition    SD2-depth/unclip    Ours     Condition    SD2-depth/unclip    Ours

Figure 14: Comparison between Uni-ControlNet and Stable Diffusion 2.1.

Table 7: CLIP score on different controllable diffusion models. The best results are in bold.

|  | Canny | MLSD | HED | Sketch | Pose | Depth | Segmentation | Style\Content |
|---|---|---|---|---|---|---|---|---|
| ControlNet | 0.2538 | 0.2481 | 0.2530 | 0.2499 | 0.2572 | 0.2558 | 0.2531 | 0.2352 |
| GLIGEN | 0.2493 | - | 0.2403 | - | 0.2534 | 0.2526 | 0.2456 | 0.2401 |
| T2I-Adapter | 0.2513 | - | - | **0.2584** | **0.2608** | 0.2559 | 0.2478 | 0.2366 |
| Ours | **0.2539** | **0.2485** | **0.2556** | 0.2542 | 0.2514 | **0.2561** | **0.2540** | **0.2402** |

**User Study:** As FID and CLIP score may be not always consistent with human preference, we further conduct user study to quantitatively compare our approach with the baseline methods ControlNet [21], GLIGEN [23] and T2I-Adapter [22]. More specifically, we carry out tests in both single and multi-condition settings, with 20 cases for each setting. Each case is evaluated based on three metrics: the quality of generated images, the match with the given text, and the alignment with the given conditions. Users should select the best one for each metric from the generated images of ControlNet, GLIGEN, T2I-Adapter, and our Uni-ControlNet. We collect responses from 20 users and analyze the total number of votes for each metric under each setting.

The results are presented in Figures 15 and 16. It can be seen that, our approach outperforms both ControlNet, GLIGEN and T2I-Adapter in the single condition setting, demonstrating a clear advantage. Additionally, in the multi-conditions setting, our approach performed significantly better than Multi-ControlNet and CoAdapter.

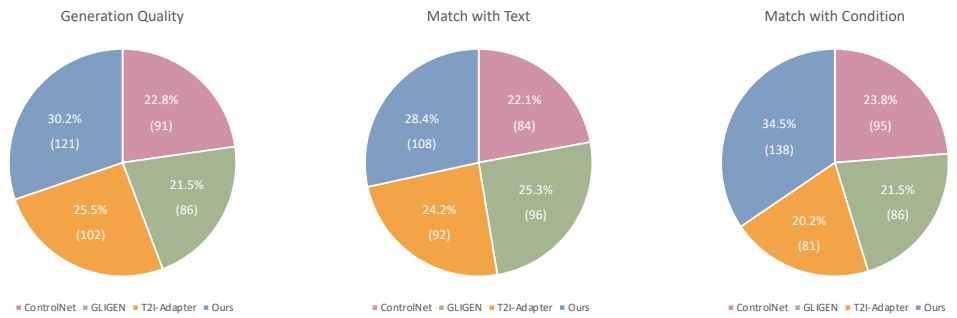

Figure 15: User study of the preference rate for the single condition setting.

## H  More Visualization Results

In this section, we present additional qualitative results. Figure 17 illustrates the results for the single-condition setting, while Figure 18 shows the results for the multi-conditions setting. Moreover, we demonstrate our performance on a more challenging case where there are four conditions, as seen in rows 7-8 of Figure 18.

Table 8: CLIP score on condition injection methods and training strategies. The best results are in bold.

|  | Canny | MLSD | HED | Sketch | Openpose | Depth | Segmentation | Content |
|---|---|---|---|---|---|---|---|---|
| Injection-S1 | 0.2513 | 0.2497 | 0.2518 | 0.2507 | 0.2527 | 0.2525 | 0.2508 | - |
| Injection-S2 | 0.2504 | **0.2506** | 0.2518 | 0.2523 | 0.2527 | 0.2544 | 0.2540 | - |
| Injection-S3 | - | - | - | - | - | - | - | **0.2502** |
| Training-S1 | 0.2506 | 0.2504 | 0.2511 | 0.2510 | 0.2529 | 0.2538 | 0.2526 | 0.2478 |
| Training-S2 | **0.2528** | 0.2504 | 0.2530 | 0.2537 | **0.2533** | 0.2547 | **0.2545** | 0.2421 |
| Ours | **0.2528** | 0.2483 | **0.2535** | **0.2539** | 0.2503 | **0.2549** | 0.2522 | 0.2420 |

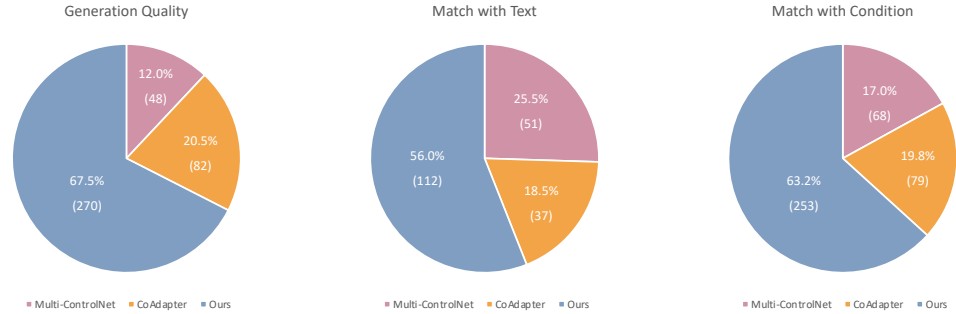

Figure 16: User study of the preference rate for the multi-conditions setting.

# I   Adapter Details

We provide the details of our proposed local control adapter and global control adapter in Figure 19, Figure 20 and Figure 21.

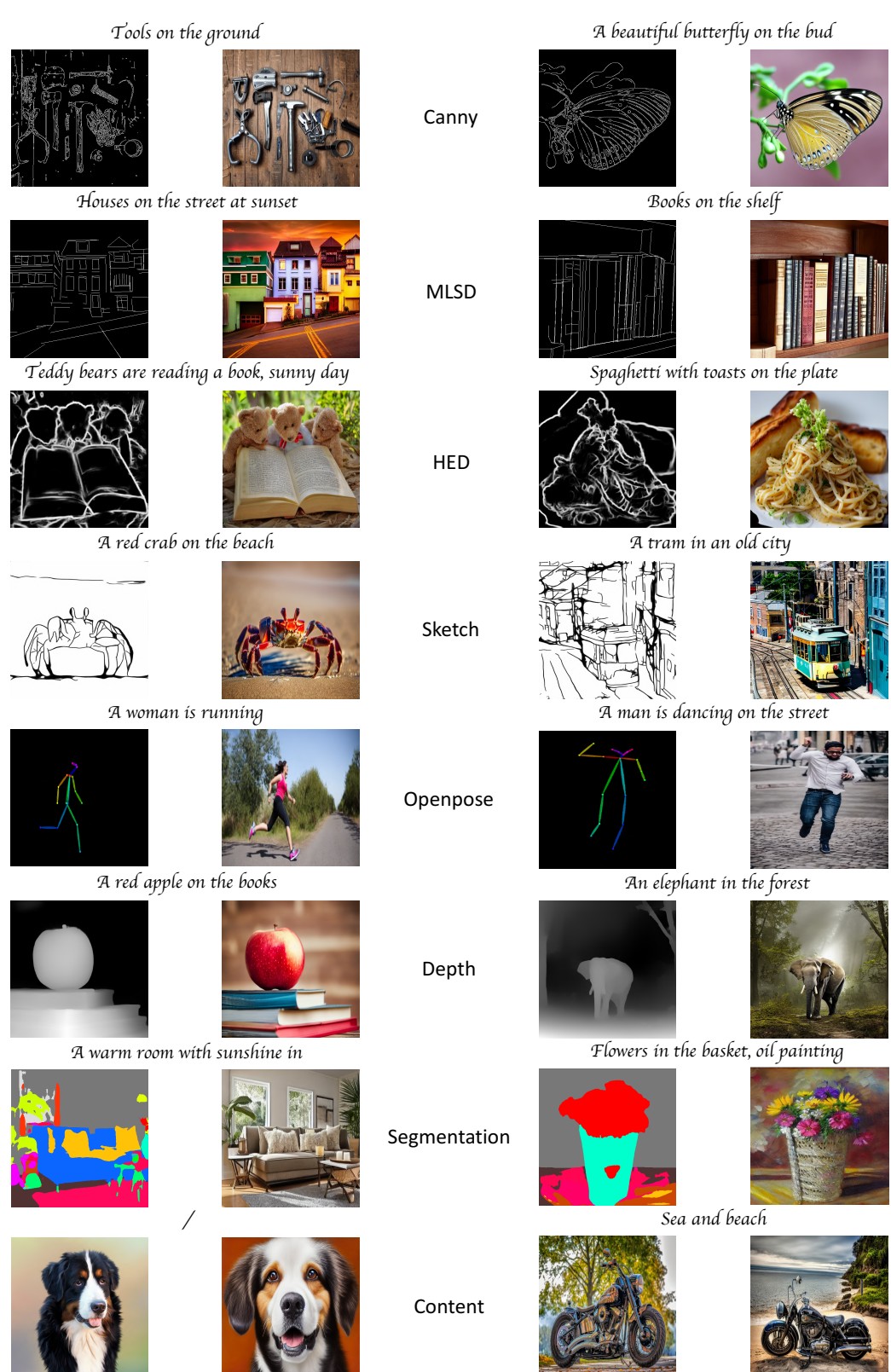

*Tools on the ground*       *A beautiful butterfly on the bud*

Canny

*Houses on the street at sunset*       *Books on the shelf*

MLSD

*Teddy bears are reading a book, sunny day*       *Spaghetti with toasts on the plate*

HED

*A red crab on the beach*       *A tram in an old city*

Sketch

*A woman is running*       *A man is dancing on the street*

Openpose

*A red apple on the books*       *An elephant in the forest*

Depth

*A warm room with sunshine in*       *Flowers in the basket, oil painting*

Segmentation

*Sea and beach*

Content

Figure 17: More visual results of Uni-ControlNet for single condition setting.

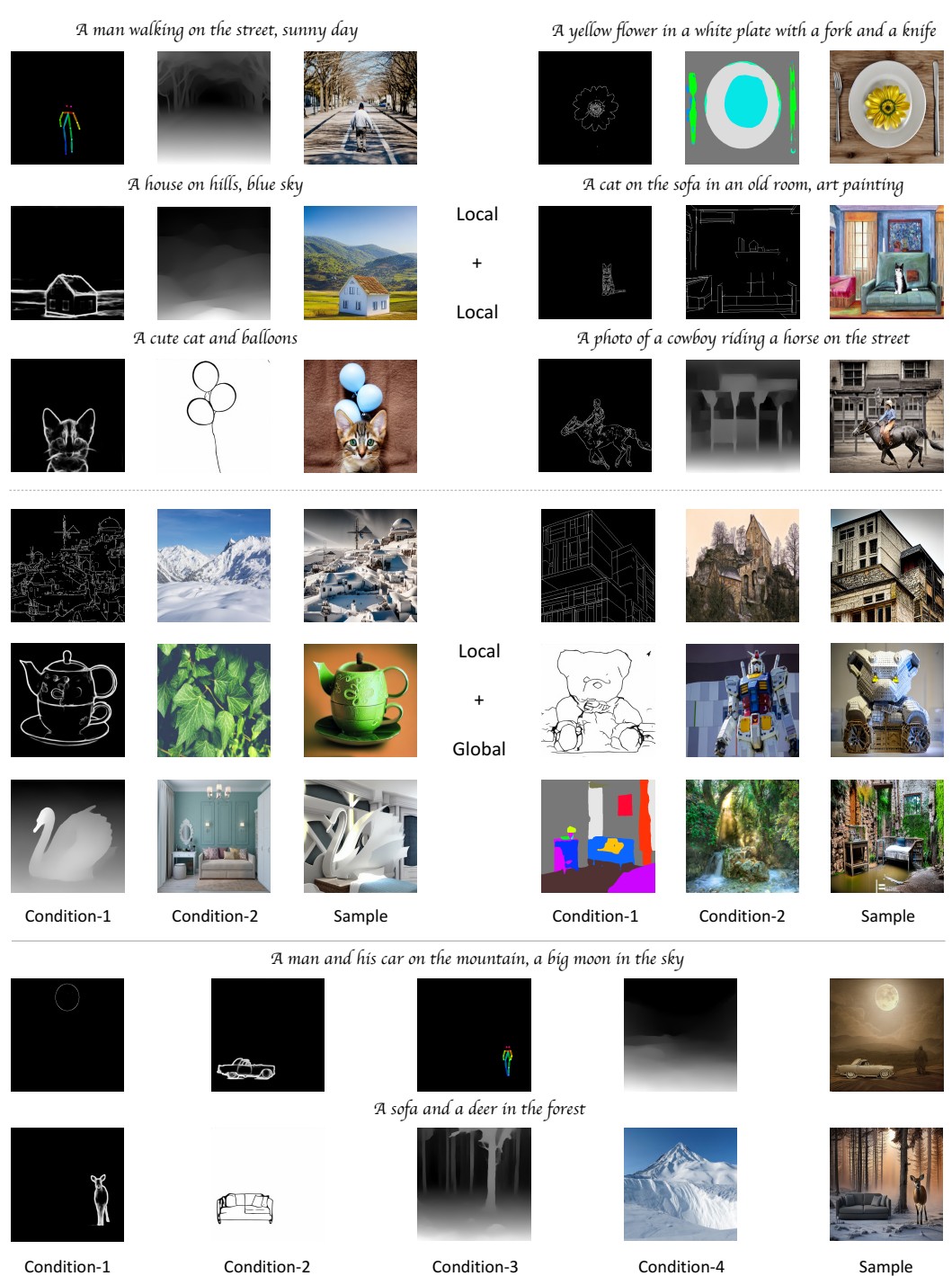

Figure 18: More visual results of Uni-ControlNet for multi-conditions setting.

## Copied Encoder

Local Conditions

Convolutional Layer
SiLU
Convolutional Layer
SiLU
Convolutional Layer
SiLU
Convolutional Layer
SiLU
Convolutional Layer
SiLU
Convolutional Layer
SiLU

Noise

Block-1 | Convolutional Layer | Zero

Block-2 | ResBlock ← FDN ← Zero | Zero
Attention Layer

Resolution: 64×64

Block-3 | ResBlock
Attention Layer | Zero
Block-4 | Down Sample | Zero

Convolutional Layer
SiLU

Block-5 | ResBlock ← FDN ← Zero
Attention Layer | Zero

Resolution: 32×32

Block-6 | ResBlock
Attention Layer | Zero
Block-7 | Down Sample | Zero

Convolutional Layer
SiLU

Block-8 | ResBlock ← FDN ← Zero
Attention Layer | Zero

Resolution: 16×16

Block-9 | ResBlock
Attention Layer | Zero
Block-10 | Down Sample | Zero

Convolutional Layer
SiLU

Block-11 | ResBlock ← FDN ← Zero | Zero
Block-12 | ResBlock | Zero

Resolution: 8×8

Copied | ResBlock
Middle | Attention Layer
Block | ResBlock | Zero

Feature Extractor

Figure 19: Details of the local control adapter.

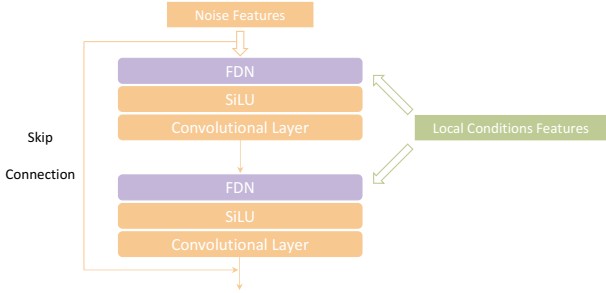

Figure 20: Details of the ResBlock with FDN in local control adapter.

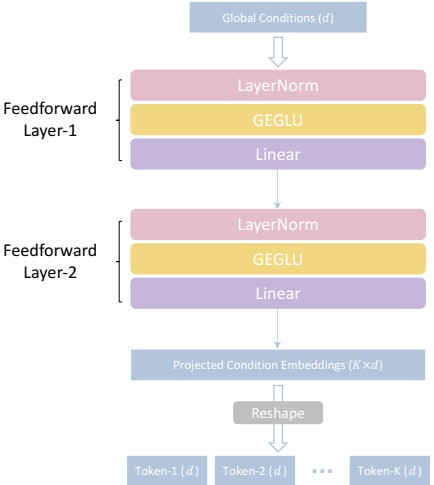

Figure 21: Details of the global control adapter. $d$ is the dimension of text token embedding and $K$ is the number of the global tokens.

