# OpenReview forum: "Uni-ControlNet: All-in-One Control to Text-to-Image Diffusion Models"
_NeurIPS.cc/2023/Conference — NeurIPS 2023 poster_

### Official Review · Reviewer_MFNW · 2023-07-01

**Soundness:** 3 good
**Presentation:** 3 good
**Contribution:** 2 fair
**Rating:** 6
**Confidence:** 5

**Summary:**

This paper presents Uni-ControlNet, a model that aims to enhance Text-to-Image diffusion techniques by allowing the concurrent use of multiple local and global controls. It only requires two additional adapters, regardless of the number of controls used, and circumvents the necessity of training from scratch. The authors assert that Uni-ControlNet performs favorably in terms of controllability, generation quality, and composability.

**Strengths:**

- The proposed method allows for the simultaneous utilization of different local and global controls within one model, making it flexible and composable.

- It eliminates the need for training from scratch, reducing costs and making it suitable for real-world deployment.

**Weaknesses:**

-	Limited Novelty and Contribution: The paper's primary contributions are its condition injection strategy and training approach. However, the condition injection strategy appears to be derived from SPADE, and the training strategy seems to be based primarily on empirical evidence, without providing much novel insight or theoretical explanation.

-	Insufficient Detail in Discussion: The description of the training strategy and the inference process, stated as major contributions, are not sufficiently clear. It remains unclear how the authors handle other conditions when only one condition is being utilized. It seems problematic to set the local conditions' values to zero with the intent of rendering them empty. Moreover, it seems that Uni-ControlNet cannot handle multiple conditions of the same type, as suggested in Figure 2 of the Supplementary Materials.

-	Incomplete Comparisons in Experiments: The experimental comparison does not seem comprehensive. For instance, it's known that Stable Diffusion 2.1 unclip can also accept CLIP image embeddings as inputs, like global condition in this paper. Additionally, ControlNet can accommodate multiple conditions. Furthermore, the quantitative results do not appear to be particularly impressive - with the method achieving the best result in only 4 out of 8 instances in Table 2, and only 2 out of 8 in Table 1.  Same for CLIP score in Supplementary Materials.


**Questions:**

-	Have you considered introducing a metric that specifically evaluates the controllability of Uni-ControlNet? FID primarily assesses image quality, but it doesn't provide a measure for controllability.

**Limitations:**

yes.

---

> ### Author Rebuttal · Authors · 2023-08-10
>
> **Thanks for your valuable comments!**
>
> **Q1. Novelty and Contribution.**
>
> **Answer**: Thanks for your comments!  Please refer to the common Q1.
>
>
> **Q2: Training strategy and inference process.**
>
> **Answer**: As mentioned in line 149 of our main paper, we concatenate all the local conditions along the channel dimension to enable simultaneous use of different conditions. During training, as stated in line 178-180 in the main paper, we adopt a predefined probability to randomly drop each condition, along with an additional probability to deliberately keep or drop all conditions. For the dropped conditions, we set the value of the corresponding input channels to 0. This simple strategy has been shown to be very effective, because the feature extractor in the local adapter can learn to understand the user intention by the supervision loss during training.
>
>
> **Q3: How to achieve composite control of 2 conditions with the same type.**
>
> **Answer**: Really good point! In real-world applications, user can actually composite two/multiple conditions of the same type easily before feeding them to the model, e.g., draw the sketch of multiple objects in one canvas. However, considering it as an interesting research point, we try one simple strategy called "Uni-Channels". Specifically, we augment the input by adding three extra condition channels. For instance, if the original inputs had 21 channels (3 for each condition, totaling 7 local conditions), with Uni-Channels, we now have 21 + 3 channels for the inputs.
>
> During training, we feed the Uni-Channels with randomly selected types of conditions of the input natural images. We observe that, as the shared extra condition channels for different condition types, Uni-Channels can perform well for two-condition composition of the same condition type. The visualization results are depicted in Figure 5 of the rebuttal PDF.
>
>
> **Q4: Comparison with Stable Diffusion 2.1.**
>
> **Answer**: Thanks for your suggestions! We compare our method with 2 Stable Diffusion models, Stable Diffusion 2 - depth ("SD2-depth") and Stable Diffusion 2 - unclip ("SD2-unclip") which support the inputs of depth map and reference image respectively. The visualization results are shown in Figure 6 in the rebuttal PDF. Additionally, we provide the quantitative results below:
>
> | FID | Depth | Content |
> |:---|:---:|:---:|
> | SD2-depth | **17.76** | / |
> | SD2-unclip | / | 24.12 |
> | Ours | 21.20| **23.98** |
>
> | CLIP score | Depth | Content |
> |:---|:---:|:---:|
> | SD2-depth | 0.2516 | / |
> | SD2-unclip | / | **0.2497** |
> | Ours | **0.2561** | 0.2402 |
>
> It is important to note that for SD2-depth and SD2-unclip, the whole model is fine-tuned to learn the depth map or the reference images instead of only fine-tuning adapters, which is the key factor contributing to their great performance. Additionally, when compared to other controllable diffusion models like T2I-Adapter and ControlNet, SD2-depth and SD2-unclip outperform them, as demonstrated in Table 2 of the main paper and Table 1 of the supplementary material.
>
>
> **Q5: Comparison with Multi-ControlNet.**
>
> **Answer**: Please refer to the common Q2.
>
>
> **Q6: Quantitative results are not particularly impressive?**
>
> **Answer**: Thanks for your valuable comments, but we respectively disagree with this point. Regarding the quantitative results of the comparison with other methods presented in Table 2 in the main paper, our approach achieves the best performance in 6 out of 8 metrics (or 4 out of 6, considering that T2I-Adapter does not consider MLSD and HED) even with one unified single model. In the ablation study, it is important to note that Training-S2 is also our method. Training-S2 involves additional joint fine-tuning after our separate fine-tuning. Therefore, in Table 3, we achieve the best results in 7 out of 8 metrics. The CLIP score provided in the supplementary file also reflects our great performance.
>
> Furthermore, our method demonstrates good results in the user study in the supplementary material, which provides reliable and straightforward indications of our superior performance from the user perception perspective.
>
>
> **Q7:Evaluation of the controllability.**
>
> **Answer**: Really great question! How to evaluation of the controllability is a common and important problem in controllable diffusion models. And we believe user study is the most accurate way to evaluate the controllability from the user perception view, and we already include this metric during the user study (provided in supplementary material). However, following your suggestion, we also try some automatic controllability evaluation metrics. Please refer to the common Q3.

---

> ### Author Response · Authors · 2023-08-14
> **Help check if questions are well addressed.**
>
> Dear Reviewer MFNW,
>
> We would like to express our appreciation for your efforts and suggestions! Could you please spare some time to check the response and see if your concerns are well addressed? We are very delighted to discuss with you and address any questions you might still have.

---

> > ### Comment · Reviewer_MFNW · 2023-08-15
> >
> > Thanks the authors for the insightful explanation! I think most of my concerns are addressed. Thus I would like to raise my rating to "weak accept".

---

> > > ### Author Response · Authors · 2023-08-16
> > > **Thanks for your comments!**
> > >
> > > Dear Reviewer MFNW,
> > >
> > > We are glad that your concerns have been well addressed. Really appreciate your prompt response and valuable feedback!

---

### Official Review · Reviewer_n8So · 2023-07-03

**Soundness:** 3 good
**Presentation:** 3 good
**Contribution:** 3 good
**Rating:** 6
**Confidence:** 4

**Summary:**

This paper proposes Uni-ControlNet that leverages lightweight local and global adapters to enable precise controls over pre-trained T2I diffusion models.

**Strengths:**

1. This paper is well written and organized.
2. The idea of local/global adapter to achieve all-in-one control is reasonable and interesting.
3. The results seem good.



**Weaknesses:**

Since this paper is easy to follow and self-consistent, I have only a few minor questions:

1. Please compare the training costs of Uni-ControlNet with those of other methods (T2IAdapter, ControlNet).
2. I notice that the different conditions are concatenated as inputs to the adapter. If we want to add other control conditions, does the adapter need to be retrained?

Overall, although this work has some limitations, I think it meets the bar of NeurIPS.

**Questions:**

See Weakness.

**Limitations:**

The authors have adequately addressed the limitations.

---

> ### Author Rebuttal · Authors · 2023-08-10
>
> **Q1: Comparing the training cost of Uni-ControlNet with other methods.**
>
> **Answer**: Great suggestions! Since the scale of the training set and training epochs varies across different methods, we present the time cost of a single training step as a measure. The reported result represents the average time cost across different conditions for each method:
>
> | | ControlNet | T2I-Adapter | Ours |
> |:---|:---:|:---:|:---:|
> | Seconds | 5.01 | **3.73** | 5.16 |
>
> The results show that both ControlNet and Our model have similar training costs, with an average of around 5 seconds per training step. In contrast, the T2I-Adapter exhibits a lower training cost, which can be attributed to the lightweight nature of a single adapter within the T2I-Adapter model.
>
> It is important to note that, regardless of the number (N) of conditions, we only need to train a unified single model. However, for ControlNet and T2I-Adapter, their training cost will increase linearly with N, as they require training a dedicated model for each specific condition.
>
>
> **Q2: Extending a trained Uni-ControlNet to newly added Conditions.**
>
> **Answer**: Super insightful question! To extend a trained Uni-ControlNet to support new conditions, we conducted an experiment in two steps for comparison & analysis purpose. Firstly, we train a local adapter specific to N conditions. Next, we introduce a new type of condition and extend the trained adapter to (N+1) conditions. The adaptation process involved modifying the input channel of the Uni-ControlNet's first convolutional layer within the feature extractor.  Then, we try to retrain the local adapter with 4 different retraining strategies (R1-4) to accommodate the new conditions:
>
> 1. Retraining the entire feature extractor (R1),
> 2. Only retraining the pre-feature extractor, which is the part that projects the condition from resolution 512 to 64 (R2),
> 3. Only retraining the first convolutional layer in the feature extractor (R3),
> 4. Without retraining, i.e., random initialization of the first convolutional layer in the feature extractor (R4).
>
> During the retraining process, we ensure that the weights of the copied encoder in the local adapter remain fixed. We utilize a training dataset of 300k samples for the retraining. We show the extension from [MLSD + HED + Sketch + OpenPose + Depth + Seg] to [MLSD + HED + Sketch + OpenPose + Depth + Seg + **Canny**]. The results of this extension process are presented in Figure 2 of the rebuttal PDF. We surprisingly observe that retraining solely the first convolutional layer in the feature extractor can already adequately enables the Uni-ControlNet to handle the newly added conditions. This is a great feature that enables our model to quickly expand to new conditions!

---

> ### Author Response · Authors · 2023-08-14
> **Is there any further questions or concerns?**
>
> Dear Reviewer n8So,
>
> We sincerely appreciate your efforts and positive feedback! Could you please help find time to review the response and see if your questions are well answered. We are very happy to discuss with you about any remaining questions you might still have.

---

> > ### Comment · Reviewer_n8So · 2023-08-15
> >
> > Thanks for the authors' rebuttal. I think my concerns have been addressed. I would like to keep the original rating (weak accept).

---

> > > ### Author Response · Authors · 2023-08-16
> > > **Thanks for your comments!**
> > >
> > > Dear Reviewer n8So, we really appreciate your valuable comments, prompt response, and recognition of our paper.

---

### Official Review · Reviewer_WHfG · 2023-07-06

**Soundness:** 2 fair
**Presentation:** 3 good
**Contribution:** 2 fair
**Rating:** 5
**Confidence:** 5

**Summary:**

This paper proposed a method to do controlleble t2i generation from a pretrained diffusion model. The main contribution is that they only have two adapters one local (e.g., edge map, keypoint etc) and one global (e.g., image). For local, they use the controlnet, but concatenate conditions as input. For global, they extract image feature and treat it as text tokens.



**Strengths:**

The idea is simple and writing is clear

**Weaknesses:**

There are several weakness for this paper:

1, the technique novelty is incremental. Although, they tried something such as SPADE like injecting information for local branch and combine image and text tokens for global etc, but they are very straightforward.

2, missing baseline GLIGEN [Li et al, CVPR, 2023] which also supports conditions studied in this paper.

3, one more weakness for this paper is missing evaluation for condition correspondence. They only reported FID as a metric, which only reflects image quality. But they should also study how well the generated images are corresponded with input conditions. For example, in GLIGEN, they use mask-rcnn to detect keypoints from generated images and compare with input keypoint, so that we can know how well the model following the input. I understand that for certain conditions such as edge map, maybe it is hard to evaluate, but as least for keypoint, semantic map, depth map, it is easy to come up with some metrics.






**Questions:**

NA

**Limitations:**

See weakness





====================================================================================

They addressed my main concern which is they only evaluated image quality, but not controllability.
I strongly encourage them to add results table in the rebuttal to their paper, thus will be served as a baseline for future controllable image generation work.

Based on this point, I am willing to raise my score despite that I feel novelty is a bit weak

---

> ### Author Rebuttal · Authors · 2023-08-10
>
> **Thanks for your valuable comments**
>
> **Q1. The technique novelty.**
>
> **Answer**: Thanks for your suggestions! Please refer to the common Q1.
>
>
> **Q2: The comparison with the GLIGEN.**
>
> **Answer**:  Thanks for your suggestions! GLIGEN is an excellent paper that introduces a model conditioned on bounding boxes with caption groundings. It also explores alternative forms of grounding, such as edge maps and pose information. While our original paper does not consider GLIGEN as a baseline due to its primary focus on bounding boxes condition and lack of support for composable control, comparing our method to GLIGEN can enhance the comprehensiveness of our study and show performance differences.
>
> To facilitate its incorporation into the final version, we directly utilize the samples presented in Figure 5 of the main paper. We showcase the qualitative comparison results in Figure 4 of the rebuttal PDF. From the results, we can easily observe some shortcomings in GLIGEN's output. For example, the detail of the deer's face is not very good, the depiction of the forest lacks realism, and the sky is missing in the case of segmentation map condition.
>
> We further conduct the quantitative comparison in terms of FID and CLIP Score:
>
> | FID | Canny | MLSD | HED | Sketch | Pose |Depth | Segmentation | Content |
> |:---|:---:|:---:|:---:|:---:|:---:|:---:|:---:|:---:|
> | GLIGEN | 24.74 | / | 28.57 | / | **24.57** | 21.46 | 27.39 | 25.12 |
> | Ours | **17.79** | **26.18** | **17.86** | **20.11** | 26.61 | **21.20** | **23.40** | **23.98** |
>
> | CLIP score | Canny | MLSD | HED | Sketch | Pose |Depth | Segmentation | Content |
> |:---|:---:|:---:|:---:|:---:|:---:|:---:|:---:|:---:|
> | GLIGEN | 0.2493 | / | 0.2403 | / | **0.2534** | 0.2526 | 0.2456 | 0.2401 |
> | Ours | **0.2539** | **0.2485** | **0.2556** | **0.2542** | 0.2514 | **0.2561** | **0.2540** | **0.2402** |
>
> To evaluate the perception quality, we further conduct a user study, following the settings in Section 3 - User Study in the supplementary material.
>
> | | Generation Quality | Match with Text | Match with Condition |
> |:---|:---:|:---:|:---:|
> | GLIGEN | 30.3% (121) | 44.2% (168) | 29.5% (118) |
> | Ours | **69.7% (279)** | **55.8% (212)** | **70.5% (282)** |
>
> We can find that our method outperforms GLIGEN in nearly all evaluations.
>
>
> **Q3: Evaluation of the controllability.**
>
> **Answer**: Really great question! How to evaluation of the controllability is a common and important problem in controllable diffusion models. And we believe user study is the most accurate way to evaluate the controllability from the user perception view, and we already include this metric during the user study (provided in supplementary material). However, following your suggestion, we also try some automatic controllability evaluation metrics. Please refer to the common Q3.

---

> ### Author Response · Authors · 2023-08-14
> **Help check the rebuttal and happy to discuss more.**
>
> Dear Reviewer WHfG,
>
> We are very grateful for your efforts and suggestions! We have addressed the concerns in the above rebuttal. Could you please help take a look and see whether your concerns are well addressed? We are very happy to discuss with you and provide further clarification for any new questions. Grateful for your effort!

---

> > ### Author Response · Authors · 2023-08-16
> > **Thanks for the updated comments**
> >
> > Dear Reviewer WHfG,
> >
> > Thanks for your prompt response and raising the score to accept. We will follow your suggestion and add all the results shown in the rebuttal into the final version.

---

### Official Review · Reviewer_RHVi · 2023-07-07

**Soundness:** 3 good
**Presentation:** 3 good
**Contribution:** 3 good
**Rating:** 6
**Confidence:** 4

**Summary:**

This paper proposes Uni-ControlNet for the simultaneous utilization of various local controls and global controls within a single model in a flexible and composable manner. This is achieved by fine-tuning of two additional adapters on top of pre-trained text-to-image diffusion models, eliminating the significant cost of training from scratch.

**Strengths:**

This paper propose a new framework that leverages lightweight adapters to enable precise controls in a single model.

**Weaknesses:**

1.	The training sets for the different models in table 2 are not the same. It raises the question of fairness in comparisons between the models.

2.	The author should compare with the simple baseline Multi-controlnet: https://huggingface.co/blog/controlnet.


**Questions:**

1.	It is unclear from the information provided whether the LAION dataset underwent any filtering, such as using OpenPose. Not all the images includes human to detect.

2.	The training time and GPU usage are not provided in the given information. These details are important for understanding the computational requirements and resource usage of the models.

3.	Is the last condition in Figure 5 image condition(global condition)? It is unclear how ControlNet implements this condition. The paper should provide a clear explanation of the condition and how it is implemented in ControlNet to ensure transparency and understanding of the model.

4.	During training, are all seven conditions of each image simultaneously inputted to the network for training, or are some conditions selectively set to empty?
5.	The evaluation of clip scores is not discussed, which is important for text-driven generation.

6. why Feature Denormalization is considered superior to using SPADE (Injection-S1) or ControlNet (Injection-S2) directly? Could you give some explanation?


**Limitations:**

yes

---

> ### Author Rebuttal · Authors · 2023-08-09
>
> **Thanks for your valuable comments!**
>
> **Q1: Fairness in comparisons.**
>
> **Answer**: It's worth noting that the training set used for the compared models is not publicly available (we already made the request by email, but no response or data not sharable). Additionally, the specific training settings for their released models are not fully disclosed as well, which makes it hard to guarantee absolute fairness in the comparison. To minimize this gap, we conducted experiments to ablate the model design between our method and other models in Section 4.3 in the main paper. Through these control-experiments, we were able to demonstrate the effectiveness of our method.
>
>
> **Q2: Comparison with Multi-ControlNet.**
>
> **Answer**: Thanks for your suggestions!  Please refer to the common Q2.
>
>
> **Q3: The construction of the training set.**
>
> **Answer**: We did not employ any filtering method for the LAION dataset, as stated in line 186 in the main paper, and used a random subset of the dataset for training. As also mentioned in line 178-180 in the main paper, we adopted a dropout strategy for conditions during training.
>
> Regarding the pose condition, it is true that not all images in the dataset include humans. To ensure that the pose condition is fully trained, we opted to not drop the pose condition during training. This means that we always keep the pose condition if it is available, allowing the model to learn the full range of pose information. In the future, if there is only a very small portion of data existing for some special conditions, advanced resampling strategies may be needed to guarantee balance and enough training.
>
>
> **Q4: Training time and GPU usage.**
>
> **Answer**: Good question! We train our model by using 64 NVIDIA Tesla 32G-V100 GPUs. We trained on 10 million data for 1 epoch, with a batch size of 192 for the local adapter and 256 for the global adapter. It took approximately 3 days to train the local adapter, and around 1.5 days to train the global adapter. However, as illustrated in Section 4.3 in our main paper, training on 1 million data for 1 epoch is sufficient to achieve great results.
>
>
> **Q5: How does ControlNet implement the content condition?**
>
> **Answer**: Yes, the last condition in Figure 5 of the main paper represents the image condition. ControlNet v1.1 achieves control through content, using an image-to-image method that differs from our embedding-to-image approach. To implement this, they first shuffle the content by remapping the image based on a random flow and then use the shuffled content to control the generation process.
>
>
> **Q6: The keep and drop of conditions during training.**
>
> **Answer**: As mentioned in line 149 in our main paper, we concatenate all the local conditions along the channel dimension to enable simultaneous use of different conditions. During training, as stated in line 178-180 in the main paper, we adopt a predefined probability to randomly drop each condition, along with an additional probability to deliberately keep or drop all conditions. For the dropped conditions, we set the value of the corresponding input channels to 0.
>
>
> **Q7: Evaluation on CLIP score.**
>
> **Answer**: Due to space constraints in the main paper, we have provided the evaluation results in terms of CLIP score in Table 1 and Table 2 of the supplementary material. In addition to the FID and CLIP score, we have also conducted a user study to evaluate the results, which can be found in Figure 3 and Figure 4 of the supplementary material. These evaluation measures provide a comprehensive assessment of the performance of our model.
>
>
> **Q8: Why is FDN better than using SPADE or ControlNet directly?**
>
> **Answer**: It is a great question! Directly using SPADE to inject conditions involves resizing the conditions to the corresponding resolutions using interpolation which we call Injection-S1 in our main paper. This direct interpolation significantly destroys condition information, leading to poor alignment with the conditions, as illustrated in Figure 7, Table 3 of the main paper. Additionally, when directly resizing the conditions and sending them to the model, the resized conditions cannot align well with the latent space where they were injected. In contrast, our FDN employs a multi-scale injection strategy that provides condition information at different levels, resulting in richer condition information. Furthermore, our feature extractor projects the conditions to the corresponding latent spaces of different layers, which allows for better alignment between the conditions and noise features.
>
> When using ControlNet or T2I-Adapter directly, which only provide condition information in the input layer of the adapter (which we refer to as Injection-S2 in our main paper), the model may lose some information of the conditions in deeper layers. This is illustrated in Figure 7 of the main paper, where Injection-S2 cannot handle composite controls effectively, and the generated samples do not align well with the combined conditions, or the conditions are not well merged.
>
> In contrast, our FDN method provides multi-scale injection, which allows for better preservation of condition information and alleviates the condition forgetting issue in deeper layers. We demonstrate that this design results in superior performance in handling composite controls within a unified framework.

---

> ### Author Response · Authors · 2023-08-14
> **Is there any more questions or concerns?**
>
> Dear Reviewer RHVi,
>
> Really thank you for your efforts and suggestions! Could you please help check our response and see whether your questions are well answered? We are very pleased to engage in a discussion with you and provide additional clarification for any new questions.

---

> > ### Comment · Reviewer_RHVi · 2023-08-16
> >
> > Thanks for the author's response. All my concerns have been well addressed. I would like to raise the score to weak accept.

---

> > > ### Author Response · Authors · 2023-08-16
> > > **Thanks for the recognition of our work**
> > >
> > > Dear Reviewer RHVi, we are glad that all your concerns have been addressed. We really appreciate your prompt response and recognition of our work.

---

### Official Review · Reviewer_WFYK · 2023-07-07

**Soundness:** 3 good
**Presentation:** 2 fair
**Contribution:** 3 good
**Rating:** 7
**Confidence:** 4

**Summary:**

The authors proposed Uni-ControlNet, a novel approach that allows for the simultaneous utilization of different local controls and global controls. It uses two additional adapters (local and global) and injects their outputs into the frozen pretrained diffusion models, and only the parameters in adapters need training.

Through both quantitative and qualitative comparisons, Uni-ControlNet demonstrates its superiority over existing methods in terms of controllability, generation quality, and composability.

**Strengths:**

1. By training with multiple conditions simultaneously, Uni-controlnet is able to perform various kinds of control with only one model.
2. Uni-controlnet only adds 2 adapters, which is efficient in both training and inference.
3. By concatenating clip image embedding with text embedding (condition), the method can control the style of generated image.

**Weaknesses:**

1. The clarification of the dataset construction is unclear. For example, is the skeleton/sketches generated by model, or manually collected? If it's automatically generated by models, the performance will be bounded by the accuracy of those models, and may suffer from distribution gaps if the control map  are painted by human during inference. Otherwise, it's very hard to anotate such a complex dataset.
2. Insufficient ablation study. As mentioned in L14, "Uni-ControlNet only necessitates a constant number (i.e., 2) of adapters, regardless of the number of local or global controls used." Is it because of the structure design? If so, a normal ControlNet with multiple controls trained together should be compared with.

**Questions:**

1. What's the inference time cost of Uni-controlNet?
2. How can we extend a trained Uni-ControlNet to other types of control? Do we need to train the adapters with all types of control together from scratch?

**Limitations:**

see weakness and questions.

---

> ### Author Rebuttal · Authors · 2023-08-09
>
> **Thanks for your valuable comments!**
>
> **Q1: How to get the training data of sketches?**
>
> **Answer**: Great question! Indeed, annotating a sketch dataset can be challenging. In our experiment, we initially obtain the HED boundary detection of an image and subsequently utilize a sketch simplification method to generate the sketches for the training samples. Although there are distribution gaps between the hand-drawn sketches and the model-generated sketches, we have observed that our model can handle hand-drawn sketches pretty well (Figure 1 of the rebuttal PDF). Please note that, how to further boost the generation quality by bridging the gap between hand-drawn sketches and model-generated sketches is beyond the scope of this paper.
>
> **Q2: Ablation study on structure design.**
>
> **Answer**: In our ablation study, we extensively investigated the impact of structure design. One aspect we explored is, as you mentioned, training a ControlNet with multiple controls simultaneously, referred to as Injection-S2 in our main paper. However, this design yielded relatively poor results, as illustrated in Figure 7 and Table 3 in the main paper, as well as Table 2 in the supplementary material.
>
> Furthermore, we conducted a comparison with another structure design called "Injection-S1", which directly utilizes SPADE. It can be seen that, Injection-S1 produced inferior results compared to both our proposed method and Injection-S2.
>
> **Q3: Inference cost.**
>
> **Answer**: We conducted a test by evaluating 100 samples for each condition and calculated the average inference time per sample:
>
> | | ControlNet | T2I-Adapter | Ours |
> |:---|:---:|:---:|:---:|
> | Seconds | 9.02 | **6.21** | 9.16 |
>
> The results indicate that the inference cost of ControlNet and Our model is approximately the same, around 9 seconds per sample on average. On the other hand, the T2I-Adapter demonstrates a faster inference time, achieving 6 seconds per sample on average. This can be attributed to the lightweight nature of a single adapter in the T2I-Adapter model.
>
> **Q4: Extending a trained Uni-ControlNet to newly added Conditions.**
>
> **Answer**: Super insightful question! To extend a trained Uni-ControlNet to support new conditions, we conducted an experiment in two steps for comparison & analysis purpose. Firstly, we train a local adapter specific to N conditions. Next, we introduce a new type of condition and extend the trained adapter to (N+1) conditions. The adaptation process involved modifying the input channel of the Uni-ControlNet's first convolutional layer within the feature extractor.  Then, we try to retrain the local adapter with 4 different retraining strategies (R1-4) to accommodate the new conditions:
>
> 1. Retraining the entire feature extractor (R1),
> 2. Only retraining the pre-feature extractor, which is the part that projects the condition from resolution 512 to 64 (R2),
> 3. Only retraining the first convolutional layer in the feature extractor (R3),
> 4. Without retraining, i.e., random initialization of the first convolutional layer in the feature extractor (R4).
>
> During the retraining process, we ensure that the weights of the copied encoder in the local adapter remain fixed. We utilize a training dataset of 300k samples for the retraining. We show the extension from [MLSD + HED + Sketch + OpenPose + Depth + Seg] to [MLSD + HED + Sketch + OpenPose + Depth + Seg + **Canny**]. The results of this extension process are presented in Figure 2 of the rebuttal PDF. We surprisingly observe that retraining solely the first convolutional layer in the feature extractor can already adequately enables the Uni-ControlNet to handle the newly added conditions. This is a great feature that enables our model to quickly expand to new conditions!

---

> ### Author Response · Authors · 2023-08-14
> **Help check whether questions are well answered.**
>
> Dear Reviewer WFYK,
>
> We would like to thank you again for your efforts and positive feedback! Could you please help find time to take a look at the response and check whether your questions are well answered. We are very happy to answer any questions you might still have.

---

> ### Comment · Reviewer_WFYK · 2023-08-17
>
> Thanks for the authors' rebuttal. I think my concerns have been addressed.

---

> > ### Author Response · Authors · 2023-08-17
> > **Thanks for your comments!**
> >
> > Dear Reviewer WFYK,
> >
> > We are glad that your concerns have been addressed! And we sincerely thank you for your valuable feedback and recognition of our paper!

---

### Author Rebuttal · Authors · 2023-08-10

**We would like to thank all the reviewers for the valuable feedback!** Here we first address some common questions.

**Q1. Re-clarification of Contribution and Novelty.**

**Answer**:  We would like to emphasize that our primary contribution is proposing a new unified controllable diffusion model that can not only handle different conditions within **one single model** but also supports composable control, as illustrated in Table 1 in the main paper. By contrast, existing methods fail to achieve this unified framework within one single model. Besides, even for those methods that support composite control, their composability is much worse than ours. Through extensive qualitative and quantitative evaluations, our method demonstrates even overall better results with a unified single model thanks to our newly proposed designs.

From the technique side, we have explored some existing techniques and found they are inadequate for achieving our goal of a unified framework. For example, the injection strategy employed by ControlNet and T2I-Adapter is insufficient, as it will suffer from information loss. Similarly, directly using SPADE will also result in poor performance, as resizing condition features through direct interpolation to low resolutions causes significant loss of condition information. Therefore, we develop a new multi-scale injection strategy through FDN to achieve better alignment between condition features and latent noise features across different layers of the local adapter, which is not explored in previous methods.

**Q2: Comparison with Multi-ControlNet.**

**Answer**: Great suggestion and thanks for the reminder! We want to explain that we missed the comparison with Multi-Controlnet because it is not included in the original ControlNet paper. However, we acknowledge the importance of comparing our method with Multi-ControlNet. To facilitate its incorporation into the final version, we utilize the samples presented in Figure 6 of the main paper and present the comparison results in Figure 3 of the rebuttal PDF.  Our observations are that Multi-ControlNet has much worse composability. For example, similar to T2I-Adapter, it misses the podium and the car in the first 2 samples in Figure 3 of the rebuttal PDF respectively. In addition, the composite generation of a local condition and a global condition is also not that good.

As Multi-ControlNet is designed for composite control, we conducted a similar user study  evaluation following the settings in Section 3 - User Study of the supplementary material. The results are presented below:

| | Generation Quality | Match with Text | Match with Condition |
|:---|:---:|:---:|:---:|
| Multi-ControlNet | 27.0% (108) | 39.5% (79) | 10.8% (43) |
| Ours | **73.0% (292)** | **60.5% (121)** | **89.2% (357)** |

It can be seen that our method demonstrates a clear advantage compared to the Multi-ControlNet on all the three metrics in the user study.

**Q3:Evaluation of the controllability.**

**Answer**: Really great question! Evaluating the controllability of different methods is a crucial aspect of the controllable diffusion models. We firmly believe that human perception provides the most effective and accurate measure in this regard, especially for the multi-condition scenarios. Therefore, we conducted a user study that encompassed both single condition and multi-condition controls, allowing participants to select the results they deemed to best match the given conditions. The results of our ablation study can be found in the supplementary material, specifically in Figure 3 and Figure 4.

While human perception serves as the most important evaluation metric, we also recognize the importance of utilizing other quantitative metrics as the auxiliary metrics to assess the controllability. Following the reviewers' suggestion, we employed the following metrics for single-condition generation:

1. SSIM (Structural Similarity) for Canny, HED, MLSD, and sketch conditions.
2. mAP (mean Average Precision) based on OKS (Object Keypoint Similarity) for pose condition.
3. MSE (Mean Squared Error) for Depth map.
4. mIoU (Mean Intersection over Union) for segmentation map.
5. CLIP score for content condition.

To calculate these metrics, we compare the extracted conditions from the natural image (the ground truth) and the corresponding generated image. We follow the settings in Section 4.2 Comparison with Existing Methods - Quantitative Comparison of the main paper and here are the results:

|  | Canny-SSIM | MLSD-SSIM | HED-SSIM | Sketch-SSIM | Pose-mAP |Depth-MSE | Segmentation-mIoU | Content-CLIP score |
|:---|:---:|:---:|:---:|:---:|:---:|:---:|:---:|:---:|
| ControlNet | 0.4828 | **0.7455** | 0.4719 | 0.3657 | 0.4359 | **87.57** | **0.4431** | 0.6765 |
| GLIGEN | 0.4226 | / | 0.4015 | / | 0.1677 | 88.22 | 0.2557 | 0.7458 |
| T2I-Adapter | 0.4422 | / | / | 0.5148 | **0.5283** | 89.82 | 0.2406 | 0.7078 |
| Ours | **0.4911** | 0.6773 | **0.5197** | **0.5923** | 0.2164 | 91.05 | 0.3160 | **0.7753** |

Our method outperforms other baseline methods in 4 out of 8 evaluation metrics. Notably, ControlNet achieves the best performance in 3 out of 8 metrics, while T2I-Adapter only excels in 1 out of 8 metrics. However, it should be noted that such methods employ different models for different conditions, allowing each model to be well-trained for its corresponding condition. In contrast, we only use a single model and achieved even overall superior results.

By utilizing both human perception (user study) and quantitative metrics, we aim to provide a comprehensive evaluation of the controllability achieved by our method and enable a thorough understanding of its performance.

---

### Decision · Program_Chairs · 2023-09-21

**Decision:**

Accept (poster)

**Comment:**

This paper was reviewed by five knowledgeable referees. The initial concerns raised by the reviewers included (1) the novelty of the proposed approach, which was perceived as incremental (WHfG, MFNW); (2) the experimental evidence, which was considered insufficient (missing ablations and comparisons, extensions to new controls unclear, fairness of the comparisons, quantitative evaluation based on FID) (WFYK, RHVi, WHfG, n8So, MFNW); (3) the dataset collection, which was not sufficiently clear (WFYK). The reviewers also raised questions about the inference cost of the proposed approach (WFYK). The rebuttal adequately addressed the reviewers' concerns by reporting conditioning-generation consistency metrics, by emphasizing the contribution, adding the requested baseline comparisons, discussing the inference cost, and clarifying the extension of the proposed approach to new controls. The reviewers acknowledge the effort that the authors put in the rebuttal and discussion and unanimously lean towards acceptance. The AC agrees with the reviewers' assessment and therefore recommends to accept. The AC encourages the authors to include the results from the rebuttal in the final version of the manuscript. The AC also encourages the authors to add results on COCO2014 validation set in the final version of the paper -- COCO2017 validation set is small and FID is sensitive to the number of samples (see results reported by GLIGEN).